# Optimizing Chain-of-Thought Reasoners via Gradient Variance Minimization in Rejection Sampling and RL

Jiarui Yao[1][*]    Yifan Hao[1]

Hanning Zhang[1]    Hanze Dong[2]    Wei Xiong[1]    Nan Jiang[1]    Tong Zhang[1]

[1]University of Illinois Urbana-Champaign
[2]Microsoft Research

## Abstract

Chain-of-thought (CoT) reasoning in large language models (LLMs) can be formalized as a latent variable problem, where the model needs to generate intermediate reasoning steps. While prior approaches such as iterative reward-ranked fine-tuning (RAFT) have relied on such formulations, they typically apply uniform inference budgets across prompts, which fails to account for variability in difficulty and convergence behavior. This work identifies the main bottleneck in CoT training as inefficient stochastic gradient estimation due to static sampling strategies. We propose GVM-RAFT, a prompt-specific Dynamic Sample Allocation Strategy designed to minimize stochastic gradient variance under a computational budget constraint. The method dynamically allocates computational resources by monitoring prompt acceptance rates and stochastic gradient norms, ensuring that the resulting gradient variance is minimized. Our theoretical analysis shows that the proposed dynamic sampling strategy leads to accelerated convergence guarantees under suitable conditions. Experiments on mathematical reasoning show that GVM-RAFT achieves a 2-4$\times$ speedup and considerable accuracy improvements over vanilla RAFT. The proposed dynamic sampling strategy is general and can be incorporated into other reinforcement learning algorithms, such as GRPO, leading to similar improvements in convergence and test accuracy.

## 1 Introduction

We consider mathematical reasoning with large language models (LLMs): given a prompt $x \in \mathcal{X}$, and aims to produce a correct final answer $z \in \mathcal{Z}$. A prevalent approach in this area is the *chain-of-thought (CoT) reasoning* (Wei et al., 2022), in which the model generates a step-by-step rationale $y \in \mathcal{Y}$ before outputting the final answer $z$. In practice, we are typically given a pre-trained and instruction fine-tuned LLM parameterized by $\theta_0$ and training samples $\{(x_i, z_i)\}$ of prompt-answer pairs. Additionally, we assume access to a verifier $r^\star(x, z) \to \{0, 1\}$ that indicates whether a predicted answer is correct or not. This is standard in practice and is particularly popular by the recent DeepSeek-R1 project (DeepSeek-AI et al., 2025), which suggests using only a symbolic verifier rather than training a neural reward model. The goal is to improve model performance by training it to generate high-quality CoT rationales that improve the final answer accuracy on unseen prompts.

We formalize CoT reasoning as a latent variable problem, treating the rationale $y$ as hidden. From this perspective, we propose a new algorithmic framework based on the expectation-maximization (EM) algorithm, which we formalize in Section 2. Prior works such as Sordoni et al. (2023); Singh et al.

---

[*]The first two authors contributed equally with random author order, detailed contributions deferred to Appendix A. Emails: {jiarui14, yifanh12, hanning5, wx13, nanjiang, tozhang}@illinois.edu, hanzedong@microsoft.com.

39th Conference on Neural Information Processing Systems (NeurIPS 2025).

(2023) have shown that this EM framework can be implemented as a variant of iterative reward-ranked fine-tuning (RAFT) (Dong et al., 2023; Touvron et al., 2023), also known as rejection sampling fine-tuning in the literature. Specifically, RAFT iteratively alternates between the E and M steps:

1. `E-step`: Prompt LLMs to generate $n$ responses per prompt, and keep responses with the highest reward only (with the correct final answers). This process can be thought to be approximating the posterior distribution of the latent variable and the evidence lower bound (ELBO). See Section 2 for a formal presentation.

2. `M-step`: Fine-tune the LLMs on the selected responses from the E-step. The fine-tuned model is used for the next-iteration E-step.

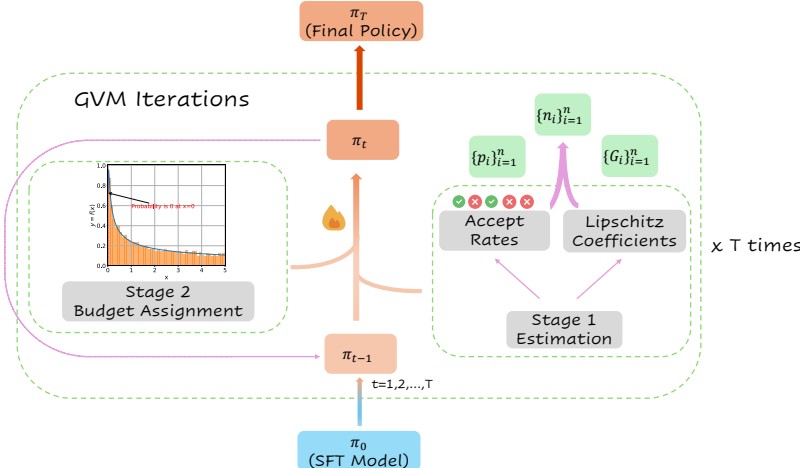

Figure 1: The demonstration of the whole pipeline for GVM. Starting from $\pi_0$, which a supervised fine-tuning (SFT) model, GVM will iteratively conduct the estimation and budget assignment process for $T$ times according to the description in Algorithm 1. Each iteration could be decomposed into two stages, and the final policy model will be returned after those iterations.

Although RAFT and its variants have been widely applied to tasks in the post-training stage of LLMs, they are primarily motivated by the idea of imitating the best-of-n sampling rather than variance reduction. As a result, theoretical understanding is often lacking. For instance, these RAFT-style approaches typically adopt a uniform sampling strategy that treats all prompts equally, ignoring differences in sample efficiency or convergence behavior

In this work, we revisit CoT reasoning under the EM framework and provide a deeper theoretical analysis. We identify the key bottleneck as the Monte Carlo estimation of the ELBO gradient during the E-step, which arises due to the intractability of going over all possible CoT rationales. Previous works mainly use the best-of-n sampling to allocate the inference budget uniformly (Sordoni et al., 2023; Singh et al., 2023), which overlook the differences in the convergence rate under different prompts. To address this, we propose a dynamic sample budget allocation strategy that adaptively assigns computational resources across prompts based on theoretical insights. This leads to a more efficient Monte Carlo estimation of the ELBO gradient. Our resulting algorithm, a refined RAFT variant with dynamic inference budget scheduling through *Gradient Variance Minimization* (GVM-RAFT), achieves substantial performance improvements and even sometimes outperforms state-of-the-art deep RL methods such as GRPO (Shao et al., 2024) in our Qwen-based case studies. In particular, the sample budget allocation schedule itself can be of independent interests and we also extend our techniques to the RL algorithm, where it also brings notable improvements. We summarize our contributions as follows:

1. We revisit the EM framework and RAFT in the context of CoT reasoning, and identify that a major limitation of current approaches lies in inefficient stochastic gradient estimation caused by uniform and static sampling strategies (i.e., best-of-n sampling), which fail to account for prompt-specific difficulty and convergence behavior.

2. Motivated by the goal of minimizing the variance of stochastic gradient, we propose a dynamic sampling strategy that adaptively allocates computational resources based on

prompt hardness and gradient norms. Our approach provides both intuitive theoretical insight and rigorous convergence guarantees, establishing a principled framework for efficient on-policy sampling under computational budget constraints.

3. We apply GVM to both RAFT++ and GRPO algorithms with real-world experiments on mathematical reasoning tasks. Our results demonstrate that the proposed approach achieves 2-4$\times$ speedup in convergence rate, with considerable improvement of the final test accuracy.

## 2 Problem Formulation and EM Framework

In this section, we formally define the problem, review existing approaches, and motivate our algorithm designs. We consider the chain-of-thought (CoT) reasoning process as:

$$x \to y \to z, \quad x \in \mathcal{X}, y \in \mathcal{Y}, z \in \mathcal{Z},$$

where $x$ is a prompt, $y$ is the intermediate CoT and $z$ is the final predicted answer. We fit the data $[x, y, z]$ within the following distribution function class:

$$\Xi := \{\mathbb{P}(y, z|x, \theta) = \mathbb{P}(y|x, \theta) \cdot \mathbb{P}(z|y, \theta) \mid \theta \in \mathbb{R}^p\}. \tag{1}$$

Our target is to learn a good model $M(\theta) \in \Xi$, which can minimize the negative log-likelihood of predicting the correct answer:

$$\mathcal{L}(\theta) = -\mathbb{E}_{x \sim d_0} \ln \mathbb{P}(z|x, \theta), \tag{2}$$

where $d_0$ is a prompt distribution and $\mathbb{P}(\cdot|\theta)$ denotes the distribution induced by the model with parameters $\theta$. While many math datasets include reference CoT rationales, we do not include these reference rationales $y$ here. This is because recent practice typically does not fine-tune LLMs on these CoT rationales after the pre-training/SFT stages (DeepSeek-AI et al., 2025; Team et al., 2025).

The process from $x$ to $z$ can be complex, making it difficult to directly estimate the distribution $P(z \mid x)$ from the observed data $[x, z]$. However, by introducing a latent intermediate variable $y$, the conditional distributions $P(y \mid x)$ and $P(z \mid y)$ are often easier to estimate, thereby simplifying the problem:

$$\mathcal{L}(\theta) = -\mathbb{E}_{x \sim d_0} \ln \sum_{y \in \mathcal{Y}} \mathbb{P}(y|x, \theta)\mathbb{P}(z|x, y, \theta).$$

Introducing the intermediate CoT $y$ naturally motivates the use of the expectation-maximization (EM) algorithm, which iteratively approximates the posterior over latent variables.

**Derivation of the EM algorithm.** We consider a training set $\mathcal{B} = \{(x_i, z_i)\}_{i=1}^m$ drawn from $d_0$ with $z_i$ being the labeled ground-truth answer to illustrate the idea. Following the standard derivation of EM algorithm, we can bound the $\mathcal{L}(\theta)$ as follows:

$$\begin{aligned}
\mathcal{L}(\theta) = -\sum_{i=1}^m \ln \mathbb{P}(z_i|x_i, \theta) &= -\sum_{i=1}^m \ln \left(\sum_{y \in \mathcal{Y}} Q_i(y)\frac{\mathbb{P}(y, z_i|x_i, \theta)}{Q_i(y)}\right) \\
&\leq -\sum_{i=1}^m \sum_{y \in \mathcal{Y}} Q_i(y) \ln \left(\frac{\mathbb{P}(y, z_i|x_i, \theta)}{Q_i(y)}\right) \\
&= \underbrace{-\sum_{i=1}^m \mathbb{E}_{y \sim Q_i(y)} \ln \mathbb{P}(y, z_i|x_i, \theta)}_{\mathcal{J}_Q(\theta)} + \sum_{i=1}^m \mathcal{H}(Q_i(y))
\end{aligned} \tag{3}$$

where $Q_i(\cdot)$ is a probability distribution over $\mathcal{Y}$ so that $Q_i(y) \geq 0$ and $\sum_{y \in \mathcal{Y}} Q_i(y) = 1$. The last inequality is from the convexity of $-\ln(\cdot)$ and Jensen's inequality (Lemma 3) and $\mathcal{H}(p) := -\mathbb{E}_{t \sim p(t)} \ln p(t)$ is the entropy.

In the E-step, our goal is to find a $Q_i(y)$ to get a sharper upper bound for $\mathcal{L}(\theta)$. In particular, the equality is reached with the posterior distribution of $y$:

$$Q_i(y) = \mathbb{P}(y|x_i, z_i, \theta) = \frac{\mathbb{P}(y, z_i|x_i, \theta)}{\mathbb{P}(z_i|x_i, \theta)} = \frac{\mathbb{P}(y|x_i, \theta) \cdot \mathbb{P}(z_i|y, \theta)}{\sum_{y \in \mathcal{Y}} \mathbb{P}(y|x_i, \theta)\mathbb{P}(z_i|y, \theta)} := \frac{\mathbb{P}(y|x_i, \theta) \cdot \mathbb{P}(z_i|y, \theta)}{Z(x_i, z_i, \theta)}, \tag{4}$$

where $Z(x_i, z_i, \theta)$ denotes the normalization constant and the second equality uses Equation (1). Here $-\mathcal{J}_Q(\theta) - \sum_{i=1}^{m} \mathcal{H}(Q_i(y))$ is referred to as the evidence lower bound (ELBO)$^*$ in the literature (Bishop and Nasrabadi, 2006; Kingma et al., 2013), and $\mathbb{P}(y|x_i, z_i, \theta)$ is the induced posterior distribution, which is not the equivalent to the autoregressive distribution of CoT when we present the LLMs with prompts and answers.

In the M-step, to minimize $\mathcal{L}(\theta)$, we can fix $Q_i(y)$ as in Equation (4) and indirectly minimize $\mathcal{J}_Q(\theta)$. Note that we do not include the entropy loss because it is a constant in the M-step since we will fix $Q_i(y)$. To summarize, the EM algorithm will alternate between the following two steps: at iteration $t$,

1. `E-step`: Update the posterior distribution of the latent CoT $Q^t$ according to Equation (4) and obtain the $\mathcal{J}_{Q^t}$ defined in Equation (3).

2. `M-step`: The goal of the M-step is to update LLMs $M(\theta_t)$ to minimize $\mathcal{J}_{Q^t}$:

$$-\frac{1}{m} \sum_{i=1}^{m} \mathbb{E}_{y \sim Q_i^t(y)} \ln \mathbb{P}(y, z_i | x_i, \theta).$$

The updated model $M(\theta_{t+1})$ is used for the next-iteration E-step.

To apply the EM algorithm, we need to sample $y \sim Q_i^t(y)$ to approximate the objective. A central challenge is that computing $Q_i^t(y)$ and $\mathcal{J}_{Q^t}(\theta)$ requires summing over all possible latent CoT reasoning $y \in \mathcal{Y}$ as we need to get the normalization constant in Equation (4). This process is usually computationally intractable. Therefore, the objective $\mathcal{J}_{Q^t}(\theta)$ must be approximated via sampling.

To approximate the posterior $Q_i(y)$, one standard approach is rejection sampling (Neal, 2003). We remark that we refer rejection sampling to the one in statistics, which is used to approximate a target distribution $Q_i(y)$ by a proposal distribution $\mathbb{P}(y|x, \theta)$, which we can sample from. In the literature of RLHF or LLM, rejection sampling is often referred to as the best-of-n sampling (Bai et al., 2022; Ouyang et al., 2022). Specifically, to approximate $Q_i(y)$ by $\mathbb{P}(y|x_i, \theta)$, the rejection sampling proceeds as follows:

1. Sample $y \sim \mathbb{P}(y|x_i, \theta)$;

2. Draw $u \sim \text{Uniform}([0, 1])$;

3. Accept $y$ if $u \leq \frac{Q_i(y)}{\mathbb{P}(y|x_i, \theta) \cdot M}$, where $M > 0$ satisfies $Q_i(y) \leq M \cdot \mathbb{P}(y|x_i, \theta)$ for all $y \in \mathcal{Y}$.

We notice that a valid choice of $M$ is $1/Z(x_i, z_i, \theta)$, leading to an accept probability of $\mathbb{P}(z_i|y, \theta)$. In mathematical reasoning, given the CoT rationale, the final predicted answer is typically of low randomness. Then, the rejection sampling in statistics aligns well with the best-of-n sampling, where we only keep all the responses with the correct answer. This eventually leads to the RAFT-type algorithms. We remark that this connection between the EM framework and the RAFT-type algorithms has been previously observed in Singh et al. (2023); Sordoni et al. (2023).

However, we argue that the current best-of-n sampling implementation is overly coarse and insufficiently faithful to the true E-step. Specifically, if a prompt $x_i$ is very difficult for current $M(\theta_t)$, the density of $\mathbb{P}(y|x_i, \theta)$ will concentrate on the wrong CoT rationales so that $\mathbb{P}(z_i|y, \theta)$ is close to zero. In this case, the accept probability will be very low and we need many samples before we can accept one valid $y$. In contrast, if our current model $M(\theta_t)$ can already output correct CoT rationale $y$ in most of the time, we will accept most of the generated responses. Eventually, with a fixed inference budget, this uniform allocation ($n$ responses per prompt) tends to bias the accepted samples toward easier prompts with higher acceptance rates.

**Notation** The true marginal negative log-likelihood is denoted as $\mathcal{L}(\theta)$. The negative ELBO at iteration $t$ is denoted as $\mathcal{L}^t$, which is equal to

$$\mathcal{L}^t(\theta) = \mathcal{J}_{Q^t}(\theta) + \sum_{i=1}^{m} \mathcal{H}(Q_i(y)).$$

Our derivation in the next section will focus on $\mathcal{J}_{Q^t}(\theta)$ since entropy term is considered to be a constant in M-step when fixing $Q^t$. We also present a notation table in Table 3 to improve readability.

---

$^*$We consider the negative log-likelihood here so it becomes an upper bound

**Algorithm 1** Meta Algorithm: GVM-EM
***
1: **Input:** Initial parameters $\theta_0$, training samples $\mathcal{D} = \{(x_i, z_i)\}_{i=1}^n$, number of epochs $T$, initial posterior $Q^0 = \mathbb{P}(\cdot \mid \theta_0)$.
2: **for** $t = 0, \dots, T$ **do**
3:     ▷ **E-step (Expectation):**
4:     Sample a set of samples $\mathcal{B}_t = \{x_i, z_i\}_{i=1}^m$. Update the posterior distribution over latent CoT rationales $Q^t(\cdot)$ using Equation (4).
5:     For each prompt $x_i$, compute the required number of samples $n_i^t$ according to (1) Theoretical Proposition 1 or (2) Practical Algorithm 2.
6:     Perform rejection sampling to obtain accepted responses $y$. Collect corresponding $(x_i, z_i, y)$ into $\mathcal{D}_i^t$, such that $y \sim Q_i^t(\cdot)$.
7:     ▷ **M-step (Maximization):**
8:     Update model parameters via gradient descent using:

$$\nabla_\theta M(\theta_t) = -\frac{1}{m} \sum_{i=1}^m \frac{1}{|\mathcal{D}_i^t|} \sum_{y_j \in \mathcal{D}_i^t} \nabla_\theta \log \mathbb{P}(y_j, z_i \mid x_i, \theta).$$

9: **end for**
10: **Output:** Final model $M(\theta_T)$.
***

## 3 Gradient Variance Minimization by Dynamic Sample Allocation

To address the limitations of best-of-n sampling, we propose a dynamic inference budget allocation strategy that adapts to the acceptance rates of rejection sampling for each prompt $x_i$. The overall meta-algorithm is presented in Algorithm 1, and in what follows, we describe the budget allocation mechanism in detail.

### 3.1 Dynamic Inference Budget Scheduling to Minimize Gradient Variance

**Unbiased gradient estimation.** We begin by formulating the true gradient at iteration $t$ under the EM objective $\mathcal{J}_{Q^t}$:

$$\nabla \mathcal{J}_{Q^t}(\theta) = -\sum_{i=1}^m \sum_{y \in \mathcal{Y}} Q_i^t(y) \nabla \ln \mathbb{P}(y, z_i | x_i, \theta) = -\sum_{i=1}^m \mathbb{E}_{y \sim Q_i^t} \nabla \ln \mathbb{P}(y, z_i | x_i, \theta), \quad (5)$$

where $Q_i^t(y) = \mathbb{P}(y | x_i, z_i, \theta_{t-1})$ is the posterior distribution of $y$. However, this distribution is intractable to compute exactly. Therefore, we approximate $Q_i^t(y)$ via rejection sampling by drawing $n_i^t$ times from current LLMs $\mathbb{P}(y | x_i, \theta_{t-1})$. This leads to the following *unbiased estimator* for Equation (5).(Detailed proof deferred to Appendix E. )

**Lemma 1** (Unbiased Gradient Estimator). *In the iteration $t$, denoting $\mathcal{D}_i^t$ as the set of accepted samples on $y$ related to $(x_i, z_i)$, we have the following unbiased gradient estimator for $\mathcal{J}_{Q^t}$:*

$$-\sum_{i=1}^m \frac{1}{n_i^t p_i^t} \sum_{y_j \in \mathcal{D}_i^t} \nabla \ln \mathbb{P}(y_j, z_i | x_i, \theta_{t-1}), \quad (6)$$

*where $p_i^t = \mathbb{E}_{y \sim \mathbb{P}(\cdot | x_i, \theta)} P(z_i | y, \theta)$ is the average accept rate of rejection sampling.*

**Variance-aware sampling allocation.** While the estimator above is unbiased, its variance can vary significantly across prompts. Prompts with low acceptance rates introduce high variance due to the small number of accepted samples. Reducing variance is crucial for efficient training with stochastic gradient, as also emphasized in prior works in statistics and optimization, including Roux et al. (2012); Johnson and Zhang (2013); Defazio et al. (2014); Chen et al. (2018). To design a more efficient sampling strategy, we analyze the variance of the gradient estimator and optimize the allocation.

**Lemma 2** (Upper Bound of Variance of Gradient Estimator ).

$$\mathbb{V}\left(\sum_{i=1}^m \frac{1}{n_i^t p_i^t} \sum_{y_j \in \mathcal{D}_i^t} \nabla\left(\ln \mathbb{P}(y_j, z_i | x_i, \theta)\right)\right) \le \sum_{i=1}^m \frac{1}{n_i^t p_i^t} \underbrace{\mathbb{E}_{y \sim Q_i^t} \left\|\nabla\left(\ln \mathbb{P}(y, z_i | x_i, \theta)\right)\right\|^2}_{G_i^2}.$$

The proof is deferred to Section E. Given a fixed total sampling budget $N$, we seek to allocate $\{n_i^t\}$ to minimize this upper bound: $\min\left\{\sum_{i=1}^m \frac{G_i^2}{p_i^t n_i^t}\right\}$, s.t. $\sum_{i=1}^m n_i^t = N$.

In practice, some prompts are totally beyond the ability of the current LLMs or cannot be evaluated by the verifier (e.g., due to some label error). This leads to extremely low acceptance rates and unstable gradient estimations. To mitigate this, we introduce a regularization term that penalizes sampling on such prompts. The revised objective becomes:

$$\min\left\{\sum_{i=1}^m \frac{1}{1+\alpha/(p_i^t)^\beta}\frac{G_i^2}{p_i^t n_i^t}\right\}, \quad \text{s.t.} \quad \sum_{i=1}^m n_i^t = N, \tag{7}$$

where $\alpha > 0, \beta \geq 2$ are hyperparameters that control the regularization strength. For example, as the accept rate $p_i^t \to 0$, the weight $(1+\alpha/(p_i^t)^\beta)^{-1} \to 0$, which leads to the fact that sample size $n_i^t \to 0$ and prevents excessive sampling on uninformative prompts. Recent studies such as Xiong et al. (2025a) have also demonstrated the critical role it plays for stable training of online RL algorithms.

Solving the regularized optimization problem in Equation (7) yields the following closed-form solution to the optimal sampling allocation:

**Proposition 1.** *The optimal number of samples allocated to each prompt is:*

$$n_i^t = N \cdot \frac{G_i/\sqrt{p_i^t + \frac{\alpha}{(p_i^t)^{\beta-1}}}}{\sum_{l=1}^n G_l/\sqrt{p_l^t + \frac{\alpha}{(p_l^t)^{\beta-1}}}} \propto \frac{G_i}{\sqrt{p_i^t + \frac{\alpha}{(p_i^t)^{\beta-1}}}}, \quad \forall i = 1, \ldots, m.$$

The proof is deferred to Appendix E.

**Remark 1.** *Accepted sample size has a lower bound as:*

$$N\sqrt{2}(\alpha(\beta-1))^{1/(2\beta)} \cdot \sum_{i=1}^m \frac{G_i}{\sum_{l=1}^m G_l} \cdot p_i^t \cdot \left(p_i^t + \frac{\alpha}{(p_i^t)^{\beta-1}}\right)^{-1/2}.$$

With Remark 1, our proposed budget scheduling method is not only efficient but also guarantees a sufficient number of accepted samples during the training process, even in the presence of informative prompts. Algorithm 2 shows a practical implementation of the GVM algorithm.

## 3.2 Theoretical Result

In this section, we present the theoretical guarantee of loss convergence. Without loss of generalization, we assume that each E-step is followed by $k$ M-steps. For the $t$-th E-step, let $b_i^r$ denote the batch size for prompt $x_i$ at the $r$-th M-step, where $kt - k < r \leq kt$, and the corresponding sample batch is denoted by $\mathcal{B}_i^r$. The upper bound loss function we construct is denoted by $\mathcal{L}_t(\theta)$:[*]

$$\mathcal{L}_t(\theta) := -\mathbb{E}_{x\sim d_0}\mathbb{E}_{y\sim Q_i^t(y)} \ln \mathbb{P}(y, z|x, \theta).$$

We take the notations below for simplification:

$$\Delta_1(k, T) := \sum_{t=1}^T \sum_{r=0}^{k-1} \mathbb{E}\|\nabla_\theta \mathcal{L}_t(\theta_{kt-k+r})\|^2 > 0, \quad \Delta_2(k, T) := \sum_{t=1}^T \mathbb{E}\|\sum_{r=0}^{k-1}\nabla\mathcal{L}_t(\theta_{kt-k+r})\|^2 > 0,$$

$$\Omega(k, T) := \sum_{t=1}^T \sum_{r=0}^{k-1} \mathbb{E}V(g_{kt-k+r}) > 0,$$

where $V(g_{kt-k+r}) = \mathbb{V}\left(-\frac{1}{m}\sum_{i=1}^m \frac{1}{b_i^{kt-k+r}}\sum_{y_j\in\mathcal{B}_i^{kt-k+r}}\nabla_\theta\left(\ln\mathbb{P}(y_j, z_i|x_i, \theta_{kt-k+r})\right)\right)$. Under mild smoothness conditions, we can derive the following result.

**Theorem 1** (Decreasing rate with smoothness condition.)**.** *Suppose* $-\ln\mathbb{P}(y, z|x, \theta)$ *is* $1/\gamma$*-smooth with respect to* $\theta$*. If* $0 < \eta \leq \gamma$*, then the proposed algorithm satisfies that*

$$\mathbb{E}\left[\mathcal{L}(\theta_{kT}) - \mathcal{L}(\theta^*)\right] - \mathbb{E}\left[\mathcal{L}(\theta_0) - \mathcal{L}(\theta^*)\right] \leq -\frac{\eta}{2}\Delta_1(k, T) + \frac{\eta^2}{2\gamma}\Omega(k, T).$$

---

[*]The expectation is taken over all sources of randomness, including the sampled data $\{\mathcal{D}_i^t\}$ and the selected batches $\{\mathcal{B}_i^r\}$.

In Theorem 1, with sufficiently large sample size, $\Omega(k, T)$ will be small enough, which ensures that the right-hand side of the inequality is negative. This guarantees the loss function decreases at each iteration. Furthermore, if the loss function exhibits convexity, we can derive the following result:

**Theorem 2** (Decreasing rate with smooth and convex condition.). *Suppose* $-\ln \mathbb{P}(y, z|x, \theta)$ *is* $1/\gamma$-*smooth and convex with respect to* $\theta$. *If* $0 < \eta \leq \gamma/2$, *then the proposed algorithm satisfies that*

$$\mathbb{E}\left[\mathcal{L}(\theta_{kT}) - \mathcal{L}(\theta^*)\right] - \mathbb{E}\left[\mathcal{L}(\theta_0) - \mathcal{L}(\theta^*)\right] \leq -\frac{\eta}{2k}\Delta_2(k, T) - \frac{\eta}{4k}\Delta_1(k, T) + \frac{\eta}{4k}\Omega(k, T).$$

In Theorem 2, the right-hand side includes an additional negative term, $-\Delta_2(k, T)$, which indicates a faster rate of decrease in the loss function. Specifically, during each E-step, the gradients across M-steps vary only slightly, this implies that:

$$\mathbb{E}\|\sum_{r=0}^{k-1} \nabla\mathcal{L}_t(\theta_{kt-k+r})\|^2 \approx k^2\mathbb{E}\|\nabla\mathcal{L}_t(\theta_{kt-k})\|_2^2 \approx k\sum_{r=0}^{k-1}\mathbb{E}\|\nabla\mathcal{L}_t(\theta_{kt-k+r})\|^2 \implies \Delta_2(k, T) \approx k\Delta_1(k, T),$$

which further induces that

$$\frac{\eta}{2k}\Delta_2(k, T) + \frac{\eta}{4k}\Delta_1(k, T) \approx \left(\frac{\eta}{2} + \frac{\eta}{4k}\right)\Delta_1(k, T) > \frac{\eta}{2}\Delta_1(k, T), \quad \frac{\eta}{4k}\Omega(k, T) < \frac{\eta^2}{2\gamma}\Omega(k, T),$$

as well as

$$-\frac{\eta}{2k}\Delta_2(k, T) - \frac{\eta}{4k}\Delta_1(k, T) + \frac{\eta}{4k}\Omega(k, T) < -\frac{\eta}{2}\Delta_1(k, T) + \frac{\eta^2}{2\gamma}\Omega(k, T).$$

All of the proofs are deferred to Appendix E.

**Guaranteed Decrease in Our Proposed Method.** From the theorems above, we know that with a sufficiently large sample size, $\Omega(k, T)$ becomes small, and the upper bound for $\mathbb{E}\left[\mathcal{L}(\theta_{kT}) - \mathcal{L}(\theta^*)\right] - \mathbb{E}\left[\mathcal{L}(\theta_0) - \mathcal{L}(\theta^*)\right]$ is strictly negative. This implies that our proposed method consistently decreases the objective loss, thereby ensuring an effective training process.

**Reduction of the variance term** $\Omega(k, T)$. According to the upper bounds derived above, a smaller $\Omega(k, T)$ leads to more efficient training. On one hand, increasing the sample size reliably reduces this term. On the other hand, under a finite budget in practice, an effective budget scheduling strategy can also reduce $\Omega(k, T)$ and thereby improve training efficiency. Our algorithm achieves this via the scheduling scheme defined in Algorithm 2.

**Trade-off in Budget Scheduling Update Frequency.** In our algorithm, the budget scheduling is updated every $k$ optimization steps. Updating more frequently (i.e., using a smaller $k$) improves training efficiency but increases computational overhead. Conversely, updating less frequently reduces computation, but as optimization proceeds, the loss gradient norm $\|\nabla_\theta\mathcal{L}_t\|_2$ tends to become small, which can slow down training. Therefore, choosing an appropriate value of $k$ requires balancing this trade-off between efficiency and computational cost.

### 3.3 Practical Implementation

**Practical implementation of GVM.** In this subsection, we describe how to implement the inference budget allocation strategy in practice, as summarized in Proposition 1, and the implementation is referred to **GVM - Gradient Variance Minimization**. Although the optimal sample sizes are of closed form, the expression involves the average accept rate $p_i^t$ and the gradient norm $G_{i,t}$, both of which depend on the posterior $Q_i^t(\cdot)$ and are therefore not available directly. Specifically, suppose that we are given the training set $\{x_i, z_i\}_{i=1}^n$ and current LLM $M(\theta)$. We can write:

$$p_i = \mathbb{E}_{y\sim\mathbb{P}(\cdot|x_i,\theta)}\mathbb{P}(z_i|y), \quad G_i = \mathbb{E}_{y\sim Q_i^t(\cdot)}\|\nabla\ln\mathbb{P}(y, z_i|x_i, \theta)\|.$$

We also use sampling to estimate these quantities. We can first generate $N'$ samples per prompt to get $\{x_i, y_i^j, z_i^j\}_{j=1}^{N'}$. Then, we can compute the following empirical estimators:

$$p_i = \frac{\sum_{j=1}^{N'}\mathbf{1}(z_i^j = z_i)}{N'}, \quad G_i = \sum_{1\leq j\leq N', z_i^j = z_i}\frac{1}{N'p_i}\|\nabla_\theta\ln\mathbb{P}(y_i^j, z_i|x_i, \theta)\|_2.$$

Then, we simply plug these empirical estimators into Proposition 1 to get the sample sizes. The entire procedure is summarized in Algorithm 2.

**GVM-RAFT++.** We implement GVM and the meta EM Algorithm 1 in a highly online fashion, building on the RAFT++ framework. In each iteration $t$, we draw a set of prompts $\{x_i, z_i\} \sim d_0$ and use the current model $\theta_{old}$ to collect $\mathcal{D}^t = \cup_i \mathcal{D}_i^t$ as the replay buffer where the inference budget allocation is determined via Algorithm 2. RAFT++ then uses these samples to compute a stochastic

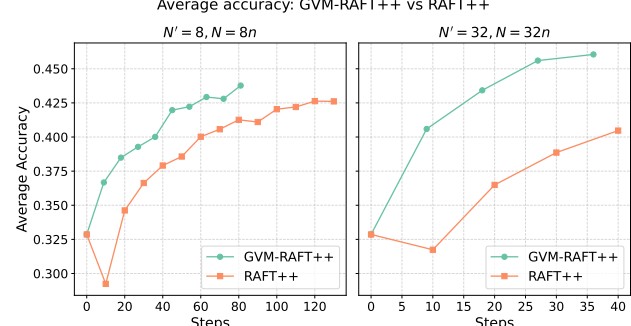

Figure 2: Average accuracy of (GVM-)RAFT++ with sample sizes 8 and 32, respectively, on Math500, Minerva Math, and Olympiad Bench, with base model Qwen2.5-Math-1.5B.

gradient estimator of the objective $\mathcal{J}_{Q^t}$. To accelerate training, we perform multiple gradient steps per iteration in a mini-batch way. This causes the model distribution to shift away from the distribution used to generate the data. To address this mismatch, RAFT++ incorporates importance sampling and clipping strategies from PPO (Schulman et al., 2017) into the original RAFT, arriving at the following loss function on the prompt-response pair $(x, a)$:

$$\mathcal{L}^{\text{RAFT++}}(\theta) = \frac{1}{|a|} \sum_{t=1}^{|a|} \Big[ \min \Big( s_t(\theta), \text{clip}(s_t(\theta), 1-\epsilon, 1+\epsilon) \Big) \Big] \cdot \mathcal{I}\big( r(x, a) = \underset{i}{\arg\max}\, r(x, a_i) \big), \quad (8)$$

where $s_t(\theta) = \frac{\pi_\theta(a_t|x, a_{1:t-1})}{\pi_{\theta_{old}}(a_t|x, a_{1:t-1})}$ and $a_t$ is the t-th token of $a$. Here, the indicator ensures that we only train on accepted responses: those approximating the posterior via rejection sampling.

**Extension to the RL algorithms.** While we focus primarily on RAFT-like algorithms, the proposed GVM strategy can be readily adapted to other RL-style fine-tuning algorithms. We focus on the GRPO, which receives significant attention recently due to its successful application to training DeepSeek-R1. Specifically, for each prompt $x$, GRPO will sample $m > 1$ responses and compute the following advantage for the t-th token of the i-th response: $A_t(x, a_i) = \frac{r_i - \text{mean}(r_1, \cdots r_m)}{\text{std}(r_1, \cdots, r_m)}$, where $r_i$ denotes the final reward of the i-th response. This leads to the following loss function for GRPO,

$$\mathcal{J}_{\text{GRPO}}(\theta) = \mathbb{E}_{x, \{y_i\}_{i=1}^m \sim \pi_{\theta_{old}}(y|x)} \Bigg[ \frac{1}{m} \sum_{i=1}^m \frac{1}{|y_i|} \sum_{t=1}^{|y_i|} \Bigg\{ \min \Bigg[ \frac{\pi_\theta(y_{i,t}|x, y_{i,<t})}{\pi_{\theta_{old}}(y_{i,t}|x, y_{i,<t})} \hat{A}_{i,t},$$

$$\text{clip}\Big( \frac{\pi_\theta(y_{i,t}|x, y_{i,<t})}{\pi_{\theta_{old}}(y_{i,t}|x, y_{i,<t})}, 1-\epsilon, 1+\epsilon \Big) \hat{A}_{i,t} \Bigg] - \beta \mathbb{D}_{\text{KL}}\big[ \pi_\theta \| \pi_{\text{ref}} \big] \Bigg\} \Bigg].$$

# 4 Experiments and Results

## 4.1 Experiments Setup

To validate the effectiveness of the proposed methods, we conduct experiments with Qwen2.5-Math-1.5B and Qwen2.5-Math-7B (Yang et al., 2024b). We focus on the mathematical reasoning task and use Math-Verify* as the verifier. For detailed experiments setup such as hyperparameters and compute resources, please refer to Appendix C.

## 4.2 Main Results

In this section, we summarize the results from integrating GVM into both RAFT++ and GRPO algorithms as a sample budget rebalancing strategy. The performance is measured by Average @ 8, which means we randomly sample 8 instances from the model with a non-zero temperature, and take

---

*https://github.com/huggingface/Math-Verify

the average accuracy as the final result. For Qwen2.5-Math-1.5B, we use a temperature of 1.0 in evaluation, while for Qwen2.5-Math-7B, we use a temperature of 0.7 as the entropy loss is higher after integrating the clip higher trick, which leads to more diverse outputs.

**GVM Improves Efficiency with Comparable Performance**  From Table 1, we could conclude that GVM could improve the final performance of both RAFT++ and GRPO when applied on Qwen2.5-Math-1.5B. For Qwen2.5-Math-7B, the final performance is comparable to baselines, under a faster convergence rate. This verifies the effectiveness of GVM using both the accept rates (standing for the hardness) and the Lipschitz coefficients (standing for the gradients) of different prompts.

Table 1: Performance of different algorithms across five benchmarks including Math500 (Hendrycks et al., 2021), Minerva Math (Lewkowycz et al., 2022), Olympiad Bench (He et al., 2024), AIME24, and AMC23. From the results, we could observe that after reweighting the sample size of prompts, GVM-RAFT++ and GVM-GRPO could outperform both vanilla RAFT++ and GRPO.

| | Method | Math500 | Minerva Math | Olympiad Bench | AIME24 | AMC23 | 5 Average |
|---|---|---|---|---|---|---|---|
| Qwen2.5 Math-1.5B | Base | 56.35 | 17.00 | 25.22 | 3.33 | 37.81 | 27.94 |
| | GRPO | 70.78 | 29.00 | 33.56 | 10.00 | 47.19 | 38.11 |
| | RAFT++ | 69.02 | 27.71 | 31.74 | 9.58 | 44.06 | 36.42 |
| | GVM-GRPO | **73.92** | **29.96** | **36.26** | **12.92** | 49.06 | **40.42** |
| | GVM-RAFT++ | 72.90 | 29.04 | 36.20 | 9.17 | **51.88** | 39.64 |
| Qwen2.5 Math-7B | Base | 42.00 | 12.82 | 19.20 | 12.92 | 30.00 | 23.39 |
| | GRPO | 81.20 | 36.03 | **44.15** | 20.83 | 63.12 | 49.07 |
| | RAFT++ | **81.68** | 35.85 | 43.83 | 20.83 | 63.12 | 49.06 |
| | GVM-GRPO | 81.55 | 36.26 | 43.56 | **22.92** | **65.00** | **49.86** |
| | GVM-RAFT++ | 81.00 | **36.67** | 43.48 | **22.92** | 61.56 | 49.13 |
| Llama-3.2 1B-Instruct | Base | 24.28 | 4.37 | 4.80 | 0.83 | 9.06 | 8.67 |
| | GRPO | **33.35** | **6.99** | **7.65** | 1.67 | 12.19 | 12.37 |
| | RAFT++ | 30.10 | 6.62 | 6.46 | 0.83 | 13.44 | 11.38 |
| | GVM-GRPO | 32.02 | 6.76 | 7.22 | **2.08** | **15.31** | **12.68** |
| | GVM-RAFT++ | 30.08 | 6.07 | 6.48 | 0.83 | 13.44 | 11.38 |
| Llama-3.2 3B-Instruct | Base | 35.62 | 9.83 | 9.09 | 2.92 | 15.00 | 14.49 |
| | GRPO | **53.40** | 19.16 | **20.67** | **8.33** | 27.50 | 25.81 |
| | RAFT++ | 47.38 | 17.69 | 17.20 | **8.33** | 29.06 | 23.93 |
| | GVM-GRPO | 51.75 | 17.74 | 19.48 | 7.50 | **33.44** | **25.98** |
| | GVM-RAFT++ | 49.05 | **20.04** | 17.87 | 6.25 | 29.06 | 23.93 |

In Figure 2, we display the step-wise performance of both RAFT++ and GVM-RAFT++ based on Qwen2.5-Math-1.5B with sample sizes per prompt of 8 and 32, respectively. GVM could enhance the convergence rate evidently, with about $2\times$ speedup for $N' = 8, N = 8n$ and $4\times$ speedup for $N' = 32, N = 32n$ measured in update steps compared to the baselines. Besides, upon convergence, GVM-RAFT++ could achieve around $1.25\%$ and $5\%$ performance gain for both configurations.

**GVM Could Be Generalized to RL Algorithms**  Though our derivation and theoretical proofs are based on EM algorithm, the sampling strategy itself could be disentangled from the pipeline and utilized solely. Take the GRPO algorithm proposed in Shao et al. (2024) as an example. Figure 6 demonstrates that with the same sample budget rebalancing strategy as in RAFT++, GVM-GRPO could perform similarly to GVM-RAFT++. This further verifies the effectiveness of GVM as a single sampling strategy compared to being deployed in the EM pipeline. Zhong et al. (2025) makes extra assumptions (Example 3.5) on the reward structure, which enables their framework's natural generalization to RL from a theoretical perspective. The similar ideas shed light on the generalization of GVM to RL algorithms, and we verified the empirical effectiveness of GVM in our experiments, indicating the success of budget reweighting through gradient variance minimization.

## 5 Conclusion, Discussion and Limitations

In summary, we propose a novel dynamic sampling and updating strategy - GVM, which could assign prompt-specific sampling budget in a fine-grained manner. The algorithm consists of two stages: a pre-sampling phase that estimates the difficulty of each instance and computes prompt-specific Lipschitz coefficients, followed by an update phase that performs parameter optimization. Our experiments have demonstrated the effectiveness of GVM, achieving faster convergence and even superior final performance under suitable settings compared to other baselines. GVM could

improve the convergence for both rejection sampling backed pipelines and then be generalized to RL algorithms like GRPO, which demonstrates the significant potential for adaptively reweighting the sampling and update budget. Finally, we also provide rigorous theoretical analysis and establish performance guarantees for this class of two-stage algorithms.

The experiments are conducted with Qwen series base models, while the effectiveness of GVM still awaits a broader verification on other base models. Besides, we believe GVM could generalize to other RL algorithms like PPO and Reinforce, while more experiments need to be performed to support the hypothesis. These could serve as the directions for possible further explorations.

## Acknowledgment

This work is partially supported by NSF grant No. 2416897 and ONR grant No. N000142512318. This research used the DeltaAI (NSF OAC-2320345) and Delta (NSF OAC-2005572) advanced computing and data resources, supported by the National Science Foundation and the State of Illinois.

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

# A  Author Contributions

This work stems from all authors' valuable contributions and close collaborations.

**JY** implements the workflow based on the algorithm pipeline; conducts preliminary experiments and verifies the effectiveness of GVM in the early stage; scales up the experiments; mainly writes the experiments-relevant sections of this paper.

**YH** initializes the project; contributes to the algorithm proposal and the clarification of the pipeline procedure; derives the theoretical analysis for GVM and establishes corresponding bounds for the algorithm; mainly writes the sections on methodology and theoretical analysis.

**HZ** devotes to the conduct of experiments; contributes to the ablation studies of the tradeoff between entropy loss and model performance; reviews related works on the techniques to mitigate fast entropy loss in the early stage of RL training and provides insightful suggestions on how to balance the tradeoff.

**HD** implements baselines and provides practical implementation suggestions; conducts a series of experiments, including baselines comparisons, GVM iterations in both RAFT++ and RL (GRPO) algorithmic settings; provides discussions and suggestions about experiment design; contributes to writing.

**WX** provides insightful advice through the project timeline on theoretical analysis, experiment design, and paper writing; contributes to the initial codebase and baseline training; mainly writes the introduction, related works, and problem formulation sections; and revises the draft version of this paper.

**NJ** and **TZ** support the work and provide computational resources, guidance, and suggestions for theoretical analysis, experiment design and paper writing.

# B  Related Work

**Iterative rejection sampling fine-tuning.**  A line of works studies the RAFT-type algorithms (iterative rejection sampling fine-tuning) (Dong et al., 2023; Touvron et al., 2023; Gulcehre et al., 2023; Yuan et al., 2023). These algorithms iteratively generate $n$ responses per prompt, discard those with incorrect answers, and fine-tune the LLM on the remaining accepted samples. Another closely related method is STaR (Zelikman et al., 2022), which also involves iterative training on self-generated CoT responses. In comparison, STaR uses greedy decoding rather than sampling, which deviates from the EM principle and suffers from severe bias across prompts of different difficulty levels (see Section 2 for details). STaR also proposes to provide the answer in the prompt to generate CoT responses for harder problems. However, our preliminary experiments suggest that LLMs often generate wrong reasoning path and output the correct answer regardless of the context. Additionally, STaR retrains from the original pre-trained model in each iteration, rather than bootstrapping from the updated model. These algorithms have been widely applied in the post-training stage of LLMs since then (Touvron et al., 2023; Xiong et al., 2025b; AI@Meta, 2024; Yang et al., 2024a). We also mention in passing that this algorithm framework can further date back to the expert iteration in the RL literature Anthony et al. (2017). Lastly, Tong et al. (2024) study inference budget allocation in the context of synthetic data construction. Their approach uses a fixed model to repeatedly sample responses until $m$ correct CoT rationales are obtained per prompt ($m$ is a hyper-parameter). While this setup also studies budget-aware sampling, they only study the offline setting and their method lacks a dynamic scheduling mechanism so cannot be applied to more general iterative online algorithms.

**Chain-of-thought reasoning under EM framework.**  Our work is also closely related with the line of work studying EM framework in CoT reasoning. Singh et al. (2023) connects RAFT with the EM algorithm under the CoT reasoning framework. We will cover such a connection in Section 2 for completeness and also discuss the limitation of the vanilla RAFT algorithm, and motivate our algorithm designs. Another related work is Sordoni et al. (2023), which proposes a gradient-free EM-based approach for prompt selection, emphasizing a different direction. Our work is also closely related to Hoffman et al. (2023), which proposes TRICE that optimizes an ELBO-inspired objective using MCMC to approximate the posterior over the latent CoT rationales. TRICE maintains a table of

prompt–CoT pairs and, in each iteration, samples a new CoT $y$ per prompt, and replace the old CoT record $y_{old}$ with $y$ if $y$ leads to correct final answer. They also propose to add a mean-zero baseline to reduce the variance of the gradient estimator (Owen and Zhou, 2000). In comparison, we design a *dynamic inference budget allocation scheduling* for a more effective estimation of the ELBO gradient by rejection sampling so the algorithm framework and techniques diverge significantly. Moreover, we implement our algorithm in a highly online and on-policy way (see Section 3.3 for details) to achieve the best performance. The lazy update of TRICE in maintaining the table of prompt-CoT pairs may not fit in this scenario. Beyond algorithmic differences, we also provide a theoretical analysis under standard conditions such as smoothness, which is missing in all these prior works. Hu et al. (2024) offers a more detailed statistical perspective on CoT reasoning as latent variable modeling. However, they mainly focus on in-context learning rather than iterative fine-tuning. Lastly, recent work Zhong et al. (2025) proposes to use a neural network to approximate the posterior distribution of latent CoT rationales and train the network by PPO. They then use the network to generate responses for the SFT or DPO algorithms, whose techniques are completely different from ours. They also provide a theoretical convergence analysis of EM algorithm under the KL-regularized Markov Decision Process (MDP) framework using the techniques from RL theory literature (Zhong et al., 2024). This type of analysis typically ignores the optimization complexity and requires to find the exact minimizer of E-step and M-step at each iteration, thus differing from our techniques and results.

**RL algorithms for LLM post training.** Reinforcement learning has played a central role in the post-training of LLMs. The most prominent example is reinforcement learning from human feedback (RLHF), which brings the first generation of Chat-GPT3.5 (Ouyang et al., 2022; Bai et al., 2022). These methods typically optimize a learned Bradley-Terry reward model (Bradley and Terry, 1952) using the Proximal Policy Optimization (PPO) algorithm (Schulman et al., 2017). To simplify the RLHF, a long line of works has proposed simplified contrastive learning objectives based on pairwise comparisons, such as Slic-HF (Zhao et al., 2023), DPO (Rafailov et al., 2023), IPO (Azar et al., 2023), GPO (Tang et al., 2024), and their online variants (Xiong et al., 2023; Dong et al., 2024; Zhang et al., 2025). After the release of OpenAI's O1 (Jaech et al., 2024) and DeepSeek's R1 (DeepSeek-AI et al., 2025), RL methods based on verifier scores, which evaluate the correctness of the final answer, have attracted significant attention in building reasoning models (Yu et al., 2025; Shi et al., 2025). This project also follows the recent convention and falls into this framework. In particular, to avoid the critic in PPO, both industry and academic researchers have revisited REINFORCE-style algorithms (Williams and Peng, 1991), including ReMax (Li et al., 2023), RLOO (Ahmadian et al., 2024; Kool et al., 2019), GRPO (Shao et al., 2024), Reinforce++ (Hu, 2025), and Reinforce-rej (Xiong et al., 2025a). We also mention in passing that the RAFT-type algorithms can also be viewed as a special variant of Reinforce-style algorithms, where reward is either $1$ or $0$. Throughout the rest of this paper, we primarily present our methods with RAFT, since it aligns most naturally with the EM objective. However, the presented dynamic inference budget scheduling can be of independent interests beyond the RAFT algorithm. We will show that integrating it into other RL-style methods, such as PPO, REINFORCE, or GRPO, can also bring notable improvements.

## C   Experiment Details

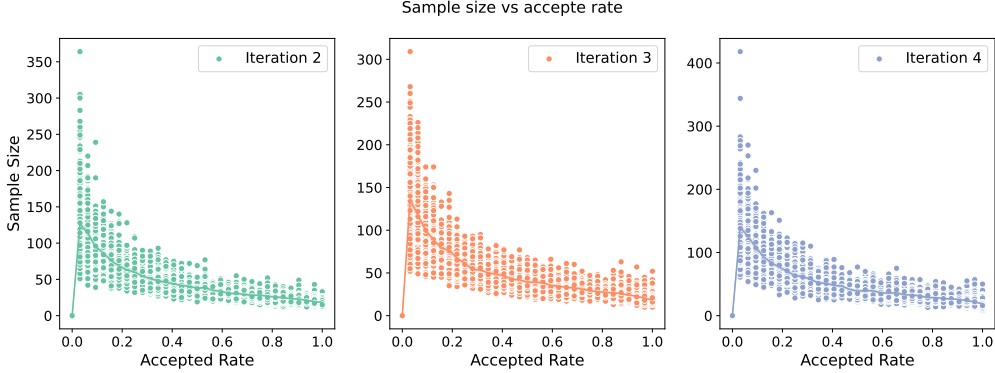

Figure 3: The assigned sample budget in GVM-RAFT++ with $N' = 32, N = 32n$ for three iterations.

**Visualization**   We present the relationship between the final sample budget and the accept rates of different prompts for GVM-RAFT++ in Figure 3. From the trend, it could be inferred that the sample budget allocated to prompts increases smoothly as the accept rates decrease. However, other affecting factors together determine the trend of the sample budget besides the hardness, where the Lipschitz coefficients measuring the contribution of the samples by the gradients play a role. If we do not take the impact of Lipschitz coefficients $\{G_i\}$ into account, then we could have the allocated sample budget to prompts with different levels of hardness measured by accept rates as shown in Figure 4.

**Hyperparameters**   We utilize verl (Sheng et al., 2024) as the training framework, and implement the RAFT++ as Xiong et al. (2025a) show that the additional importance sampling and clipping improve over the vanilla RAFT (Dong et al., 2023). For each iteration, we use a prompt batch size of 1024, and use a mini-batch size 256 for gradient update. The max prompt length is set to be 1024, and the models are allowed to generate at most 3072 tokens so that they do not exceed the context window of 4096 tokens. There is no warmup stage and the learning rate is chosen to be a constant $1e - 6$. For the training dataset, we use the Numina-Math (LI et al., 2024). For simplicity of implementation, we update the posterior distribution (i.e., the GVM in Algorithm 2) every 10 steps. In other words, we will compute the inference buget allocation for the next 10 M-steps and take 10 M-steps before entering another E-step. The 10 M-steps and the 1 E-step are together referred to as one iteration of GVM-RAFT++. We also experiment with the clip higher (Yu et al., 2025) to use an asymmetric clip threshold design. Specifically, we clip the importance sampling ratio $\pi_\theta/\pi_{\theta_{old}}$ to the range $(1 - \varepsilon_{low}, 1 + \varepsilon_{high})$, where $\varepsilon_{low} = 0.2$, and $\varepsilon_{high} = 0.28, 0.4$ then $0.35$ for RAFT++ and GVM-RAFT++ respectively. We omit this technique in GRPO training as we observe it leads to worse performance in our experiments.

To understand how to choose the hyperparameters $\alpha$ and $\beta$, we first simplify the question and omit the impacts of gradients here. Then the sample weight of function $f(x) = \frac{1}{\sqrt{x + \frac{\alpha}{x^{\beta-1}}}}$ with different choices of $\alpha$ and $\beta$ is visualized as the figure below. From Figure 4, we could see that in order to assign more sample weights on problems with lower accept rates, we should choose both $\alpha$ and $\beta$ as low as possible within a reasonable region. Therefore, in our experiments, we fix $\alpha = 0.001$ and $\beta = 2.0$. For all the hyperparameters, please refer to Table 2.

Table 2: Full hyperparameters.

| Parameter | Value |
|---|---|
| $\alpha$ | $1e - 3$ |
| $\beta$ | 2 |
| batch size | 1024 |
| mini batch size | 256 |
| max prompt length | 1024 |
| max response length | 3072 |
| learning rate | $1e - 6$ |
| KL loss coefficient | 0.001 |

As for the choices of $\alpha$ and $\beta$ is GVM stage two sample budget assignment, we visualize the effects of both parameters in Figure 4. From the figure, we could see that when $\beta$ is fixed, the smaller $\alpha$ is, the more sample budget will be allocated to harder prompts. Similarly, when $\alpha$ is fixed, the smaller $\beta$ is, the more sample budget will be allocated to harder prompts.

To analyze the impacts of which layers are used in the gradients calculation and the reduction methods of gradients, we conduct a series of experiments using Qwen2-math-7B-Instruct (Yang et al., 2024a). For the layers from which we retrieve the gradients, we choose the top one, `lm_head`, and the bottom one, `embed_tokens`, as the activated layers for loss backpropagation and gradients calculation. For gradients reduction, we compare both summation and average by sequence length. From the results shown in Figure 5, we could see that different kinds of layers for gradient calculation do not make a huge difference on the final sample sizes, while the summation of gradients biases the sample sizes to problems with lower accept rates, which we speculate the correct answers for problems with lower accept rates could steer the model more than easier problems. Therefore, in our experiments, we choose summation as the method for gradients reduction. According to the relationship between the sample sizes and accept rates. The relationship between gradients and accept rates, it could be

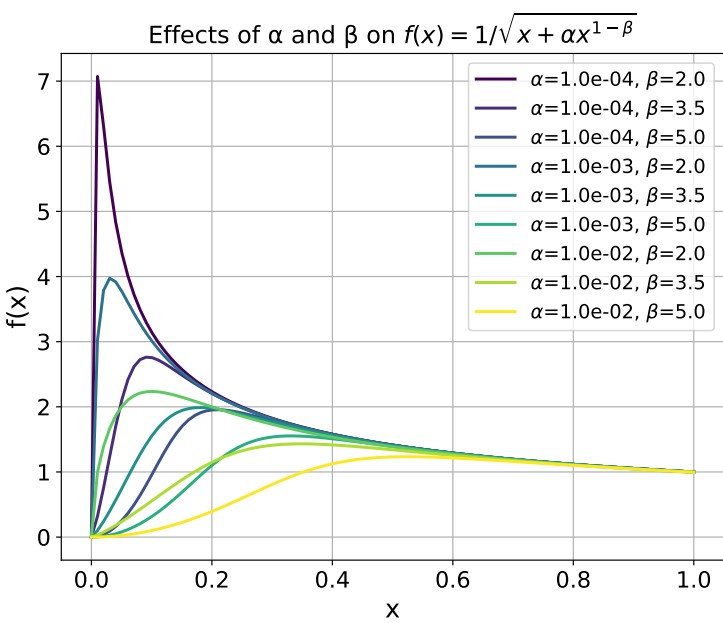

Figure 4: Visualization of different choices of $\alpha$ and $\beta$ for $f(x) = 1/\sqrt{x + \alpha x^{1-\beta}}$.

inferred that in a high level, the gradients are larger for prompts with lower accept rates if we use the average reduction. At the same time, it is not always the case, as prompts with higher accept rates may also have large gradients.

In the second stage of GVM, to make the total sample budget exactly $N$, we sort the prompts according to their accept rates in descending order, and add more samples to the prompts with lower accept rates. In our experiments, for GVM, each iteration consists of nine steps of update as the amount of training data per iteration is slightly less than $10 \times 1024 = 10240$, which is the amount of data needed for ten steps of update with train batch size 1024.

For the implementation of GVM-GRPO, we divide the sample budget calculated in the second stage by a fixed size (for example, 4 in our experiments), and copy the prompts multiple times by $n_i/4$. Then we set the rollout number to 4 during the training procedure.

For compute resources, we mainly conduct the experiments on NVIDIA RTX A6000 and H100 GPUs, and each iteration in GVM typically takes 90 minutes with sample sizes $N' = 8, N = 8n$ on a 4×H100 GPU server.

## C.1 Ablation Study

**GVM Balances the Sampling Budget in a Stable Way** Figure 9 displays the performance trend concerning the training steps, with each training step corresponding to exactly $1024 \times 4$ samples, which means that the definition of "step" here may not be mapped to the actual training step. Take the case $N' = 16, N = 16n$ for example, under this setting, each iteration is optimized over 1024 prompts, with a total sampling budget $1024 \times 16 = 4 \times (1024 \times 4)$, therefore corresponding to 4 steps in the figure. Though increasing the number of samples in the first stage, $N'$, could make more prompts be accepted at least once, thus leading to more useful prompts during the training procedure, the convergence rate does not change evidently. This means GVM could possibly estimate the difficulties of the prompts with a relatively small sampling budget to a sufficient threshold, and we could choose small $N'$ and $N$ in realistic applications.

**Existing RL training paradigms with Qwen-2.5-Math-7B suffers from distribution collapse and worse pass@n.** In Figure 6, we observe that GVM consistently improves the convergence rate for both RAFT++ and GRPO, and across both the Qwen-2.5-Math-1.5B and 7B models. However, we also find that (1) GVM yields larger gains with the 1.5B model than with the 7B model; and (2)

GVM is more effective when combined with GRPO than with RAFT++. To better understand these observations—as well as the dynamics of GVM—we conduct further ablation studies. Specifically, one can interpret vanilla RAFT++ as imitating a fixed best-of-$n$ policy. In contrast, GVM-RAFT++ adaptively selects a best-of-$n_i$ policy for each prompt $x_i$, based on its difficulty and gradient signal. GRPO (REINFORCE-based) further incorporates unlearning from negative samples. Intuitively, all these methods are influenced by the quality of the underlying best-of-$n_i$ policy. To examine this effect, we plot the pass@n curves of different checkpoints below.

We plot the pass@n curves of GVM-RAFT++ using both the Qwen-2.5-Math-1.5B and 7B models in Figure 7. As shown in the figures, the pass@n curve (for $n \geq 4$) remains relatively stable across different training iterations for the 1.5B model. In contrast, the performance consistently degrades with the 7B model. Notably, the GVM-RAFT++ 7B model at iteration 10 underperforms even the 1.5B model when $n \geq 40$. Since the imitation target gets worse and worse during the training, the GVM-RAFT++ does not improve the final test performance with the 7B model. We also mention in passing that it is expected that the current RL training paradigm cannot improve the pass@n at large $n$, a trend also observed in prior work (Shao et al., 2024; Xiong et al., 2024). However, our results show that the pass@n actually gets worse with the Qwen-2.5-Math-7B model.

We note that the observed performance difference essentially arises from existing RAFT++ or GRPO training paradigm, rather than the proposed computational resource allocation strategy (i.e., GVM). In Figure 8, we plot the pass@n curves for both the vanilla RAFT++ and vanilla GRPO. As shown in the first two plots, RAFT++ without GVM already exhibits the same trend: the 1.5B model maintains a stable pass@n curve, whereas the 7B model shows a clear degradation as training progresses. The middle plot further indicates that the "clip higher" technique helps mitigate this decline hence improving the final test performance. Finally, GVM still improves GRPO performance with the 7B model, likely because its pass@n curve deteriorates more slowly than RAFT++. This is due to the use of negative samples in GRPO (Xiong et al., 2025a).

In addition, we observe that for Qwen2.5-Math-1.5B, when $N' = 8$, the training rewards of GVM-RAFT++ are higher than RAFT++, and when $N' = 16$, the training rewards of GVM-RAFT++ are approximately the same as RAFT++, while when $N' = 32$, the training rewards are lower than RAFT++. This implies that using a larger $N'$ for GVM will bias the sample budget to harder prompts, which may be because with a larger $N'$, harder problems are more likely to be solved at least once compared to when $N'$ is smaller.

# D  Notation Table

All of the notations are listed in Table 3.

Table 3: The table of notations used in this paper.

| Notation | Description |
|---|---|
| $x, y, z$ | Prompt, CoT rationale, and predicted answer |
| $\mathcal{X}, \mathcal{Y}, \mathcal{Z}$ | the space of prompt, CoT rationale, and predicted answer |
| $m$ | the number of training samples $(x_i, z_i)$ |
| $Q_i^t(y)$ | the posterior distribution $\mathbb{P}(y|x_i, z_i, \theta)$ on the $t$-th E-step |
| $\mathcal{D}_i^t$ | the collected sample set for $(x_i, z_i, y)$ on the $t$-th E-step |
| $n_i^t$ | the size of scheduling budget for prompt $x_i$ on the $t$-th E-step |
| $p_i^t$ | the average accept rate for prompt $x_i$ on the $t$-th E-step |
| $N$ | the whole size of sampling budget |
| $(\alpha, \beta)$ | tuning parameters in budget scheduling |
| $\mathcal{B}_i^r$ | the sample batch corresponding to $x_i$ on the $r$-th M-step |
| $b_r^i$ | the batch size of $\mathcal{B}_i^r$ |
| $n_i^t$ | required number of samples for prompt $x_i$ on the $t$-th E-step |
| $k$ | the number of M-step after each E-step |
| $\theta_{kt-k+r}$ | the model parameter on the $r$-th M-step within the $t$-th E-step |
| $\eta$ | learning rate |
| $\mathcal{L}_t(\theta)$ | upper bound loss function on the $t$-th E-step |

# E   More Theoretical Results and Missing Proofs

*Proof of Lemma 1 .* Taking the expectation of Equation (6), we obtain:

$$
\mathbb{E}\left[-\sum_{i=1}^{m}\frac{1}{n_i^t p_i^t}\sum_{y_j\in\mathcal{D}_i^t}\nabla\ln\mathbb{P}(y_j,z_i|x_i,\theta_{t-1})\right]
$$

$$
= -\sum_{i=1}^{m}\frac{1}{n_i^t p_i^t}\sum_{i=1}^{n_i^t}\mathbb{E}_{y_j\sim\mathbb{P}(\cdot|x_i,\theta_{t-1})}\left[\mathbf{1}(\text{accept } y_j)\cdot\nabla\ln\mathbb{P}(y_j,z_i|x_i,\theta_{t-1})\right]
$$

$$
= -\sum_{i=1}^{m}\frac{1}{n_i^t p_i^t}\sum_{j=1}^{n_i^t}p_i^t\mathbb{E}_{y_j\sim Q_i^t}\nabla\ln\mathbb{P}(y_j,z_i|x_i,\theta_{t-1})
$$

$$
= -\sum_{i=1}^{m}\mathbb{E}_{y\sim Q_i^t}\nabla\ln\mathbb{P}(y,z_i|x_i,\theta_{t-1}),
$$

which is the same as Equation (5). $\qquad\square$

*Proof of Lemma 2 .*

$$
\mathbb{V}\left(\sum_{i=1}^{m}\frac{1}{n_i^t p_i^t}\sum_{y_j\in\mathcal{D}_i^t}\nabla\left(\ln\mathbb{P}(y_j,z_i|x_i,\theta)\right)\right)
$$

$$
= \sum_{i=1}^{m}\frac{1}{(n_i^t p_i^t)^2}\sum_{j=1}^{n_i^t}\mathbb{V}_{y_j\sim\mathbb{P}(\cdot|x_i,\theta_{t-1})}\left(\mathbf{1}(\text{accept } y_j)\nabla\left(\ln\mathbb{P}(y_j,z_i|x_i,\theta)\right)\right)
$$

$$
\leq \sum_{i=1}^{m}\frac{1}{(n_i^t p_i^t)^2}\sum_{j=1}^{n_i^t}\mathbb{E}_{y_j\sim\mathbb{P}(\cdot|x_i,\theta_{t-1})}\left\|\mathbf{1}(\text{accept } y_j)\nabla\left(\ln\mathbb{P}(y_j,z_i|x_i,\theta)\right)\right\|^2
$$

$$
= \sum_{i=1}^{m}\frac{1}{(n_i^t p_i^t)^2}\sum_{j=1}^{n_i^t}p_i^t\mathbb{E}_{y_j\sim Q_i^t}\left\|\nabla\left(\ln\mathbb{P}(y_j,z_i|x_i,\theta)\right)\right\|^2
$$

$$
= \sum_{i=1}^{m}\frac{1}{n_i^t p_i^t}\underbrace{\mathbb{E}_{y\sim Q_i^t}\left\|\nabla\left(\ln\mathbb{P}(y,z_i|x_i,\theta)\right)\right\|^2}_{G_i^2}.
$$

$\qquad\square$

If we consider further strongly convex loss function, there will be:

**Theorem 3** (Decreasing rate with smooth and strongly convex condition.). *Suppose* $-\ln\mathbb{P}(y,z|x,\theta)$ *is* $1/\gamma$-*smooth and* $H$-*strongly convex with respect to* $\theta$. *If* $0<\eta\leq\min\{\gamma/2,1/(4kH)\}$, *then the proposed algorithm satisfies that*

$$
\mathbb{E}\left[\mathcal{L}(\theta_{kT})-\mathcal{L}(\theta^*)\right]-\mathbb{E}\left[\mathcal{L}(\theta_0)-\mathcal{L}(\theta^*)\right]\leq-\frac{H\eta^2}{2k}\Delta_3(k,T)-\frac{\eta}{4k}\Delta_1(k,T)+\frac{\eta}{8k}\Omega(k,T),
$$

*where*

$$
\Delta_3(k,T):=\sum_{t=1}^{T}\sum_{r=1}^{k}\mathbb{E}\|\sum_{m=0}^{r-1}\nabla\mathcal{L}_t(\theta_{kt-k+m})\|^2>0.
$$

*Proof of Proposition 1.* Based on the objective function in Equation (7), we can define the Lagrange objective function as

$$
\mathcal{G}(\{n_i^t\},\lambda):=\sum_{i=1}^{m}\frac{1}{1+\alpha/(p_i^t)^\beta}\frac{G_i^2}{p_i^t n_i^t}-\lambda(\sum_{i=1}^{m}n_i^t-N).
$$

Taking derivative on the variables, we could obtain that

$$\frac{\partial \mathcal{G}}{\partial n_i^t} = -\frac{2}{1 + \alpha/(p_i^t)^\beta} \frac{G_i^2}{p_i^t (n_i^t)^2} - \lambda n_i^t, \quad i \in [m],$$

$$\frac{\partial \mathcal{G}}{\partial \lambda} = N - \sum_{i=1}^m n_i^t.$$

To achieve the quations as

$$\frac{\partial \mathcal{G}}{\partial n_i^t} = 0, \quad \frac{\partial \mathcal{G}}{\partial \lambda} = 0,$$

we can obtain the result as

$$n_i^t = N \cdot \frac{G_i / \sqrt{p_i^t + \frac{\alpha}{(p_i^t)^{\beta-1}}}}{\sum_{l=1}^n G_l / \sqrt{p_l^t + \frac{\alpha}{(p_l^t)^{\beta-1}}}} \propto \frac{G_i}{\sqrt{p_i^t + \frac{\alpha}{(p_i^t)^{\beta-1}}}}, \quad \forall i \in [m],$$

which finishes the proof.

$\square$

*Proof for Theorem 1.* To simplify the analysis, we denote

$$g_{kt-k+r} = -\frac{1}{m} \sum_{i=1}^m \frac{1}{b_i^{kt-k+r}} \sum_{y_j \in \mathcal{B}_i^{kt-k+r}} \nabla_\theta \left( \ln \mathbb{P}(y_j, z_i | x_i, \theta_{kt-k+r}) \right).$$

With the fact that $-\ln \mathbb{P}(y, z | x, \theta_t)$ are $1/\gamma$-smooth, we can obtain

$$\mathbb{E}\left[ \mathcal{L}_t(\theta_{kt-k+r+1}) - \mathcal{L}_t(\theta_{kt-k+r}) \right] \leq -\eta \mathbb{E} \|\nabla_\theta \mathcal{L}_t(\theta_{kt-k+r})\|^2 + \frac{\eta^2}{2\gamma} \mathbb{E} \|g_{kt-k+r}\|^2,$$

for any $r = 0, 1, \dots, k-1$ on iter-$t$. Summing the above inequality over all $r$, and with the fact that

$$\mathbb{E}\mathcal{L}(\theta_{kt}) \leq \mathbb{E}\mathcal{L}_t(\theta_{kt}) \leq \mathbb{E}\mathcal{L}_t(\theta_{kt-k}) = \mathbb{E}\mathcal{L}(\theta_{kt-k}),$$

we have

$$\mathbb{E}\left[ \mathcal{L}(\theta_{kt}) - \mathcal{L}(\theta_{kt-k}) \right] \leq -\eta \sum_{r=0}^{k-1} \mathbb{E}\|\nabla_\theta \mathcal{L}_t(\theta_{kt-k+r})\|^2 + \sum_{r=0}^{k-1} \frac{\eta^2}{2\gamma} \mathbb{E}\|g_{kt-k+r}\|^2$$

$$= -\eta \sum_{r=0}^{k-1} \mathbb{E}\|\nabla_\theta \mathcal{L}_t(\theta_{kt-k+r})\|^2 + \sum_{r=0}^{k-1} \frac{\eta^2}{2\gamma} \left( \mathbb{E}V\left(g_{kt-k+r}\right) + \mathbb{E}\|\nabla_\theta \mathcal{L}_t(\theta_{kt-k+r})\|^2 \right)$$

$$\leq -\frac{\eta}{2} \sum_{t=0}^{k-1} \mathbb{E}\|\nabla_\theta \mathcal{L}_t(\theta_{kt-k+r})\|^2 + \sum_{r=0}^{k-1} \frac{\eta^2}{2\gamma} \mathbb{E}V\left(g_{kt-k+r}\right),$$

where the last inequality is due to $\eta/\gamma \leq 1$. Further summing $t$ over $1, \dots, T$, we can obtain that

$$\mathbb{E}\left[ \mathcal{L}(\theta_{kT}) - \mathcal{L}(\theta_0) \right] \leq -\frac{\eta}{2} \sum_{t=1}^T \sum_{r=0}^{k-1} \mathbb{E}\|\nabla_\theta \mathcal{L}_t(\theta_{kt-k+r})\|^2 + \sum_{t=1}^T \sum_{r=0}^{k-1} \frac{\eta^2}{2\gamma} \mathbb{E}V\left(g_{kt-k+r}\right).$$

$\square$

*Proof for Theorem 2.* The proof techniques are mainly from Zhao and Zhang (2014). To be specific, for each iteration $t$, we denote that

$$\delta_{kt-k+r} = \left\langle -\frac{1}{m} \sum_{i=1}^m \frac{1}{b_i^{kt-k+r}} \sum_{y_j \in \mathcal{B}_i^{kt-k+r}} \nabla_\theta \left( \ln \mathbb{P}(y_j, z_i | x_i, \theta_{kt-k+r}) \right), \theta_{kt-k+r} - \theta_{kt-k} \right\rangle$$

$$+ \frac{1}{m} \sum_{i=1}^m \frac{1}{b_i^{kt-k+r}} \sum_{y_j \in \mathcal{B}_i^{kt-k+r}} \left( \ln \mathbb{P}(y_j, z_i | x_i, \theta_{kt-k+r}) - \ln \mathbb{P}(y_j, z_i | x_i, \theta_{kt-k}) \right),$$

which implies that

$$\mathbb{E}\delta_{kt-k+r} = \mathbb{E}\langle \nabla \mathcal{L}_t(\theta_{kt-k+r}), \theta_{kt-k+r} - \theta_{kt-k}\rangle - \mathbb{E}\mathcal{L}_t(\theta_{kt-k+r}) + \mathbb{E}\mathcal{L}_t(\theta_{kt-k}) \geq 0.$$

Based on these notations, we can obtain that

$$\|\theta_{kt-k+r} - \theta_{kt-k}\|^2 - \|\theta_{kt-k+r+1} - \theta_{kt-k}\|^2 = 2\langle \eta g_{kt-k+r}, \theta_{kt-k+r} - \theta_{kt-k}\rangle - \|\eta g_{kt-k+r}\|^2$$

$$= 2\eta\delta_{kt-k+r} - \frac{2\eta}{m}\sum_{i=1}^{m}\frac{1}{b_i^{kt-k+r}}\sum_{y_j\in\mathcal{B}_i^{kt-k+r}}(\ln\mathbb{P}(y_j, z_i|x_i, \theta_{kt-k+r}) - \ln\mathbb{P}(y_j, z_i|x_i, \theta_{kt-k})) - \|\eta g_{kt-k+r}\|^2,$$

taking expectation of both sides, we have

$$\mathbb{E}\|\theta_{kt-k+r} - \theta_{kt-k}\|^2 - \mathbb{E}\|\theta_{kt-k+r+1} - \theta_{kt-k}\|^2 \geq 2\eta\left(\mathbb{E}\mathcal{L}_t(\theta_{kt-k+r}) - \mathbb{E}\mathcal{L}_t(\theta_{kt-k})\right) - \eta^2\mathbb{E}\|g_{kt-k+r}\|^2.$$

Summing over $r = 0, 1, \ldots, k-1$, we have

$$\sum_{r=0}^{k-1}\mathbb{E}\left(\mathcal{L}_t(\theta_{kt-k+r}) - \mathcal{L}_t(\theta_{kt-k})\right) \leq -\frac{1}{2\eta}\mathbb{E}\|\theta_{kt} - \theta_{kt-k}\|^2 + \frac{\eta}{2}\sum_{r=0}^{k-1}\mathbb{E}\|g_{kt-k+r}\|^2.$$

Also, considering the fact obtained in previous analysis as:

$$\sum_{r=0}^{k-1}\mathbb{E}\left[\mathcal{L}_t(\theta_{kt-k+r+1}) - \mathcal{L}_t(\theta_{kt-k+r})\right] \leq -\eta\sum_{r=0}^{k-1}\mathbb{E}\|\nabla_\theta\mathcal{L}_t(\theta_{kt-k+r})\|^2 + \sum_{r=0}^{k-1}\frac{\eta^2}{2\gamma}\mathbb{E}\|g_{kt-k+r}\|^2,$$

we can combine the two inequalities above, and obtain that

$$\mathbb{E}\left[\mathcal{L}(\theta_{kt}) - \mathcal{L}(\theta_{kt-k})\right] \leq \mathbb{E}\left[\mathcal{L}_t(\theta_{kt}) - \mathcal{L}_t(\theta_{kt-k})\right] \leq \frac{1}{k}\sum_{r=0}^{k-1}\mathbb{E}\left[\mathcal{L}_t(\theta_{kt-k+r+1}) - \mathcal{L}_t(\theta_{kt-k})\right]$$

$$\leq -\frac{1}{2\eta k}\mathbb{E}\|\theta_{kt} - \theta_{kt-k}\|^2 - \frac{\eta}{k}\sum_{r=0}^{k-1}\mathbb{E}\|\nabla_\theta\mathcal{L}_t(\theta_{kt-k+r})\|^2 + \frac{1}{k}\left(\frac{\eta}{2} + \frac{\eta^2}{2\gamma}\right)\sum_{r=0}^{k-1}\mathbb{E}\|g_{kt-k+r}\|^2$$

$$= -\frac{\eta}{2k}\mathbb{E}\|\sum_{r=0}^{k-1}\nabla\mathcal{L}_t(\theta_{kt-k+r})\|^2 - \frac{1}{k}\left(\frac{\eta}{2} - \frac{\eta^2}{2\gamma}\right)\sum_{r=0}^{k-1}\mathbb{E}\|\nabla_\theta\mathcal{L}_t(\theta_{kt-k+r})\|^2 + \frac{\eta^2}{2\gamma k}\sum_{r=0}^{k-1}\mathbb{E}V\left(g_{kt-k+r}\right)^2$$

$$\leq -\frac{\eta}{2k}\mathbb{E}\|\sum_{r=0}^{k-1}\nabla\mathcal{L}_t(\theta_{kt-k+r})\|^2 - \frac{\eta}{4k}\sum_{r=0}^{k-1}\mathbb{E}\|\nabla_\theta\mathcal{L}_t(\theta_{kt-k+r})\|^2 + \frac{\eta}{4k}\sum_{r=0}^{k-1}\mathbb{E}V\left(g_{kt-k+r}\right)^2,$$

where the last inequality is from $\eta/\gamma \leq 1/2$.

Summing over $t = 1, \ldots, T$, we have

$$\mathbb{E}\left[\mathcal{L}(\theta_{kT}) - \mathcal{L}(\theta_0)\right] \leq -\sum_{t=1}^{T}\frac{\eta}{2k}\mathbb{E}\|\sum_{r=0}^{k-1}\nabla\mathcal{L}_t(\theta_{kt-k+r})\|^2 - \sum_{t=1}^{T}\frac{\eta}{4k}\sum_{r=0}^{k-1}\mathbb{E}\|\nabla_\theta\mathcal{L}_t(\theta_{kt-k+r})\|^2$$

$$+ \sum_{t=1}^{T}\frac{\eta}{4k}\sum_{r=0}^{k-1}\mathbb{E}V\left(g_{kt-k+r}\right)^2.$$

$\square$

*Proof for Theorem 3.* For each iteration $t$, we denote that

$$\delta_{kt-k+r} = \left\langle -\frac{1}{m}\sum_{i=1}^{m}\frac{1}{b_i^{kt-k+r}}\sum_{y_j\in\mathcal{B}_i^{kt-k+r}}\nabla_\theta\left(\ln\mathbb{P}(y_j, z_i|x_i, \theta_{kt-k+r})\right), \theta_t - \theta_{kt-k}\right\rangle$$

$$+ \frac{1}{m}\sum_{i=1}^{m}\frac{1}{b_i^{kt-k+r}}\sum_{y_j\in\mathcal{B}_i^{kt-k+r}}\left(\ln\mathbb{P}(y_j, z_i|x_i, \theta_{kt-k+r}) - \ln\mathbb{P}(y_j, z_i|x_i, \theta_{kt-k})\right)$$

$$- \frac{H}{2}\|\theta_t - \theta_{kt-k}\|_2^2,$$

which implies that

$$\mathbb{E}\delta_{kt-k+r} = \langle \nabla \mathcal{L}_t(\theta_{kt-k+r}), \theta_{kt-k+r} - \theta_{kt-k} \rangle - \mathcal{L}_t(\theta_{kt-k+r}) + \mathcal{L}_t(\theta_{kt-k}) - \frac{H}{2}\|\theta_{kt-k+r} - \theta_{kt-k}\|_2^2 \geq 0.$$

Based on these notations, we can obtain that

$$\|\theta_{kt-k+r} - \theta_{kt-k}\|^2 - \|\theta_{kt-k+r+1} - \theta_{kt-k}\|^2$$

$$= 2\langle \eta_s g_{kt-k+r}, \theta_{kt-k+r} - \theta_{kt-k} \rangle - \|\eta_s g_{kt-k+r}\|^2$$

$$= 2\eta\delta_{kt-k+r} - \frac{2\eta}{m}\left[\sum_{i=1}^{m}\frac{1}{b_i^{kt-k+r}}\sum_{y_j \in \mathcal{B}_i^{kt-k+r}}(\ln \mathbb{P}(y_j, z_i | x_i, \theta_{kt-k+r}) - \ln \mathbb{P}(y_j, z_i | x_i, \theta_{kt-k})) + \frac{H}{2}\|\theta_{kt-k+r} - \theta_{kt-k}\|^2\right]$$

$$- \|\eta g_{kt-k+r}\|^2,$$

taking expectation of both sides, we have

$$\mathbb{E}\|\theta_{kt-k+r} - \theta_{kt-k}\|^2 - \mathbb{E}\|\theta_{kt-k+r+1} - \theta_{kt-k}\|^2 \geq 2\eta\left(\mathbb{E}\mathcal{L}_t(\theta_{kt-k+r}) - \mathcal{L}_t(\theta_{kt-k}) + \frac{H}{2}\|\theta_{kt-k+r} - \theta_{kt-k}\|^2\right)$$

$$- \eta^2\mathbb{E}\|g_{kt-k+r}\|^2.$$

Summing over $r = 0, 1, \ldots, k-1$, we have

$$\sum_{r=0}^{k-1}\mathbb{E}\left(\mathcal{L}_t(\theta_{kt-k+r}) - \mathcal{L}_t(\theta_{kt-k})\right) \leq -\frac{1}{2\eta}\mathbb{E}\|\theta_{kt-k} - \theta_{kt-k}\|^2 - \frac{H}{2}\sum_{r=0}^{k-1}\mathbb{E}\|\theta_{kt-k+r} - \theta_{kt-k}\|^2 + \frac{\eta}{2}\sum_{r=0}^{k-1}\mathbb{E}\|g_{kt-k+r}\|^2.$$

Also, considering the fact obtained in previous analysis as:

$$\sum_{r=0}^{k-1}\mathbb{E}\left[\mathcal{L}_t(\theta_{kt-k+r+1}) - \mathcal{L}_t(\theta_{kt-k+r})\right] \leq -\eta\sum_{r=0}^{k-1}\mathbb{E}\|\nabla_\theta\mathcal{L}_t(\theta_{kt-k+r})\|^2 + \sum_{r=0}^{k-1}\frac{\eta^2}{2\gamma}\mathbb{E}\|g_{kt-k+r}\|^2,$$

we can combine the two inequalities above, and obtain that

$$\mathbb{E}\left[\mathcal{L}(\theta_{kt}) - \mathcal{L}(\theta_{kt-k})\right] \leq \mathbb{E}\left[\mathcal{L}_t(\theta_{kt}) - \mathcal{L}_t(\theta_{kt-k})\right] \leq \frac{1}{k}\sum_{r=0}^{k-1}\mathbb{E}\left[\mathcal{L}_t(\theta_{kt-k+r+1}) - \mathcal{L}_t(\theta_{kt-k})\right]$$

$$\leq -\frac{1}{2\eta k}\mathbb{E}\|\theta_{kt} - \theta_{kt-k}\|^2 - \frac{H}{2k}\sum_{r=0}^{k-1}\mathbb{E}\|\theta_{kt-k+r} - \theta_{kt-k}\|^2 - \frac{\eta}{k}\sum_{r=0}^{k-1}\mathbb{E}\|\nabla_\theta\mathcal{L}_t(\theta_{kt-k+r})\|^2$$

$$+ \frac{1}{k}\left(\frac{\eta}{2} + \frac{\eta^2}{2\gamma}\right)\sum_{r=0}^{k-1}\mathbb{E}\|g_{kt-k+r}\|^2$$

$$= -\frac{\eta}{2k}\mathbb{E}\|\sum_{r=0}^{k-1}\nabla\mathcal{L}_t(\theta_{kt-k+r})\|^2 - \frac{1}{k}\left(\frac{\eta}{2} - \frac{\eta^2}{2\gamma}\right)\sum_{r=0}^{k-1}\mathbb{E}\|\nabla_\theta\mathcal{L}_t(\theta_{kt-k+r})\|^2 + \frac{\eta^2}{2\gamma k}\sum_{r=0}^{k-1}\mathbb{E}V(g_{kt-k+r})$$

$$- \frac{H\eta^2}{2k}\sum_{r=1}^{k}\mathbb{E}\|\sum_{m=0}^{r-1}\nabla\mathcal{L}_t(\theta_{kt-k+m})\|^2 - \frac{H\eta^2}{2k}\sum_{r=1}^{k}\sum_{m=0}^{r-1}\mathbb{E}\mathbb{V}(g_{kt-k+m})$$

$$\leq -\frac{\eta}{2k}\mathbb{E}\|\sum_{r=0}^{k-1}\nabla\mathcal{L}_t(\theta_{kt-k+r})\|^2 - \frac{H\eta^2}{2k}\sum_{r=1}^{k}\mathbb{E}\|\sum_{m=0}^{r-1}\nabla\mathcal{L}_t(\theta_{kt-k+m})\|^2 - \frac{\eta}{4k}\sum_{r=0}^{k-1}\mathbb{E}\|\nabla_\theta\mathcal{L}_t(\theta_{kt-k+r})\|^2$$

$$+ \frac{\eta}{8k}\sum_{r=0}^{k-1}\mathbb{E}V(g_{kt-k+r}),$$

where the last inequality is from $\eta/\gamma \leq 1/2$.

Summing over $t = 1, \ldots, T$, we have

$$\mathbb{E}\left[\mathcal{L}(\theta_{kT}) - \mathcal{L}(\theta^*)\right]$$

$$\leq \mathbb{E}\left[\mathcal{L}(\theta_0) - \mathcal{L}(\theta^*)\right] - \sum_{t=1}^{T}\frac{H\eta^2}{2k}\sum_{r=1}^{k}\mathbb{E}\|\sum_{m=0}^{r-1}\nabla\mathcal{L}_t(\theta_{kt-k+m})\|^2 - \sum_{t=1}^{T}\frac{\eta}{4k}\sum_{r=0}^{k-1}\mathbb{E}\|\nabla_\theta\mathcal{L}_t(\theta_{kt-k+r})\|^2$$

$$+ \sum_{t=1}^{T}\frac{\eta}{8k}\sum_{r=0}^{k-1}\mathbb{E}V(g_{kt-k+r}).$$

$\square$

### E.1 Discussion on sample size

To be more specific, to achieve a negative upper bound, the sample size on each batch should satisfy that:

$$b_i \geq \frac{n_i^{-1} \sum_{j=1}^{n_i} \|\nabla_\theta \ln \mathbb{P}(y_j, z_i | x_i, \theta)\|_2^2}{\|n_i^{-1} \sum_{j=1}^{n_i} \nabla_\theta \ln \mathbb{P}(y_j, z_i | x_i, \theta)\|_2^2}.$$

## F  Technical Lemmas

**Lemma 3** (Jensen's Inequality). *Suppose that $\phi(w)$ is a convex function on $\Omega$. Consider $w_1, \cdots, w_m \in \Omega$, and non-negative numbers $\alpha_1, \cdots, \alpha_m \in \mathbb{R}$ so that $\sum_{i=1}^m \alpha_i = 1$. Then,*

$$\phi(\sum_{i=1}^m \alpha_i w_i) \leq \sum_{i=1}^m \alpha_i \phi(w_i).$$

*More generally, let $p$ be a probability measure on $\Omega$, then $\phi(\mathbb{E}_{w \sim p} w) \leq \mathbb{E}_{w \sim p} \phi(w)$. In particular, since $\|\cdot\|$ is convex (by triangle inequality of the norm), we know that*

$$\|\mathbb{E}z\| \leq \mathbb{E}\|z\|.$$

*Proof.* See Proposition A.9 of Zhang (2023) for a proof. $\square$

## G  More on GVM Algorithms

Here we present a practical version of GVM in Algorithm 2.

---

**Algorithm 2** GVM: Practical Implementation

---

1: **Input:** Model parameter $\theta$, training samples $\{x_i, z_i\}_{i=1}^n$, total inference budget $N$, parameter estimation sample size $N'$, penalty parameter $\{\alpha, \beta\}$.
2: **for** $i = 1, \ldots, m$ **do**
3:    For each prompt $x_i$, sample $N'$ times to get $\{x_i, y_i^j, z_i^j\}_{j=1}^{N'}$.
4:    Estimate the accept rate $p_i^t$ and Lipschitz bounds $G_i$ on each prompt $x_i$ as

$$p_i = \frac{\sum_{j=1}^{N'} \mathbf{1}(z_i^j = z_i)}{N'},$$

$$G_i = \sum_{1 \leq j \leq N', z_i^j = z_i} \frac{1}{N' p_i} \|\nabla_\theta \ln \mathbb{P}(y_i^j, z_i | x_i, \theta)\|_2.$$

5:    Calculate sample size $\{n_i\}$ as:

$$n_i = N \cdot \frac{G_i / \sqrt{p_i + \frac{\alpha}{(p_i)^{\beta-1}}}}{\sum_{l=1}^n G_l / \sqrt{p_l + \frac{\alpha}{(p_l)^{\beta-1}}}}.$$

6: **end for**
7: **Output:** $\{n_i\}_{i=1}^m$.

---

A general algorithm pipeline for rejection sampling is displayed in Algorithm 3, while in practice, we directly use the outcome-based reward signal to decide whether a prompt will be chosen or not.

---

**Algorithm 3** Rejection sampling

---

1: **Input:** probability with current model $\{\mathbb{P}(y|x_i, \theta_{t-1}), \mathbb{P}(z_i|y, \theta_{t-1})\}$, auxiliary probability $q(y|x_i, z_i)$, sample size $n_i^t$, accept rate $p_i^t$, empty set $\mathcal{D}$
2: **for** $j = 1, \ldots, n_i^t$ **do**
3:     Obtain a sample point $y_j$ from the distribution $q(y|x_i, z_i)$.
4:     Sample a random point $u_j \sim \mathcal{U}(0, 1)$.
5:     Calculate $Q_i(y_j)$ as:

$$Q_i(y_j) = \frac{\mathbb{P}(y_j|x_i, \theta_{t-1})\mathbb{P}(z_i|y_j, \theta_{t-1})}{\sum_y \mathbb{P}(y|x_i, \theta_{t-1})\mathbb{P}(z_i|y, \theta_{t-1}) / \sum_y \sum_z \mathbb{P}(y|x_i, \theta_{t-1})\mathbb{P}(z|y, \theta_{t-1})}$$

6:     **if** $u_j \leq \frac{p_i^t Q_i(y_j)}{q(y_j|x_i, z_i)}$ **then**
7:         Enlarge $\mathcal{D} = \mathcal{D} \cup \{y_j\}$.
8:     **end if**
9: **end for**
10: **Output:** set $\mathcal{D}$.

---

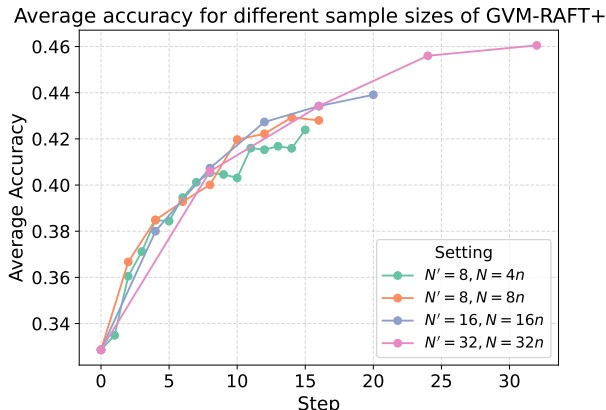

Figure 9: The average accuracy of GVM-RAFT++ with different sample sizes for both stage one and stage two on Math500, Minerva Math and Olympiad Bench. From the trend, though the number of sample sizes in both stage one and stage two increases, the convergence rate stays almost the same.

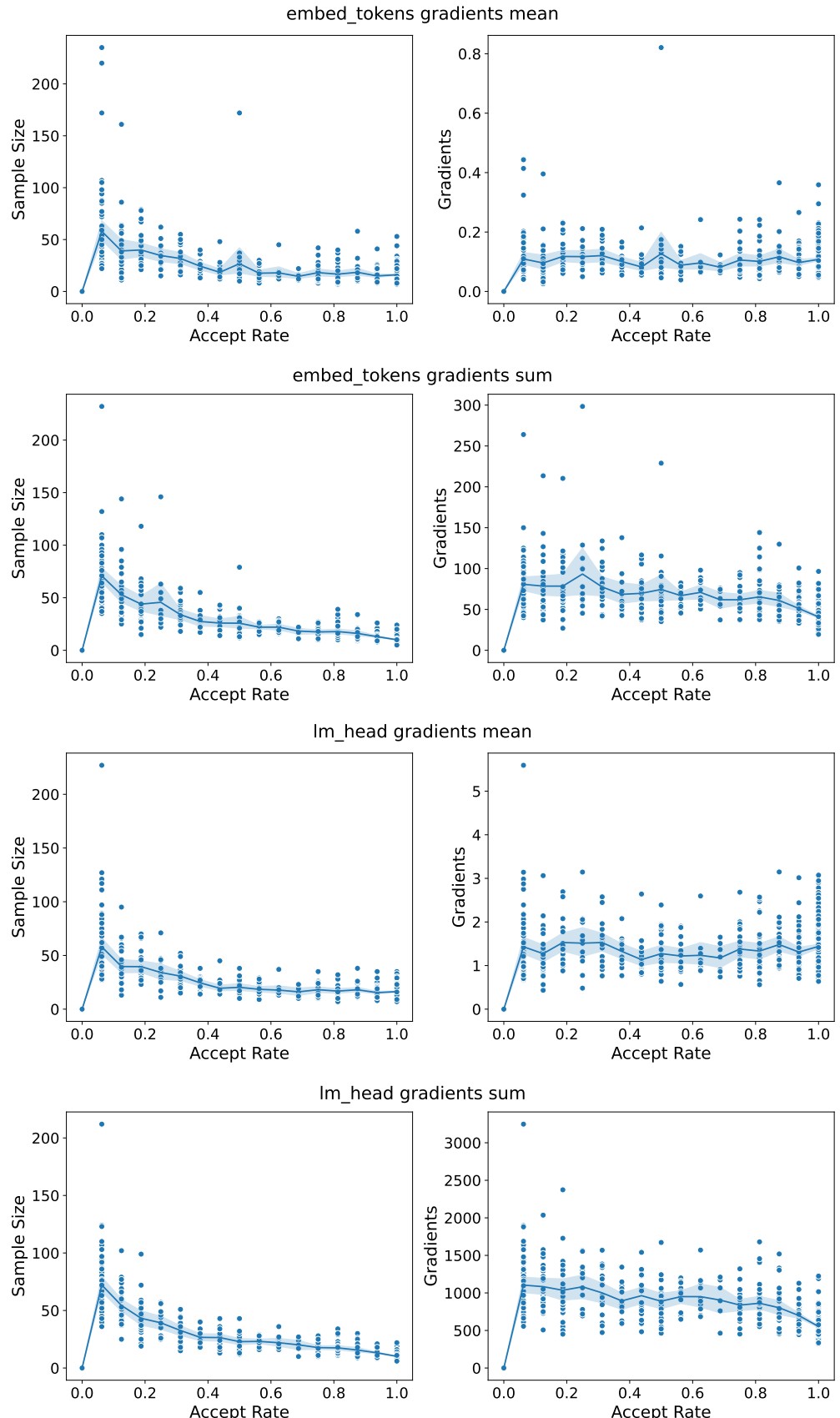

Figure 5: Comparison among different choices of the layers for gradient calculation and the methods of gradients reduction on Qwen2.5-Math-7B.

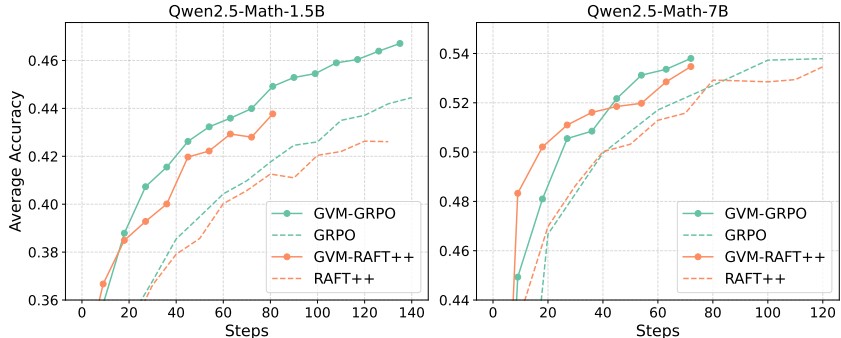

Figure 6: The average accuracy of RAFT++, GVM-RAFT++ and GRPO, GVM-GRPO with $N' = 8, N = 8n$ respectively on Math500, Minerva Math and Olympiad Bench. Applying the GVM sample strategy to RL algorithms like GRPO achieves similar results to GVM-RAFT++ compared to vanilla GRPO. For vanilla RAFT++ and GRPO, the rollout number per prompt is set to 8 as well.

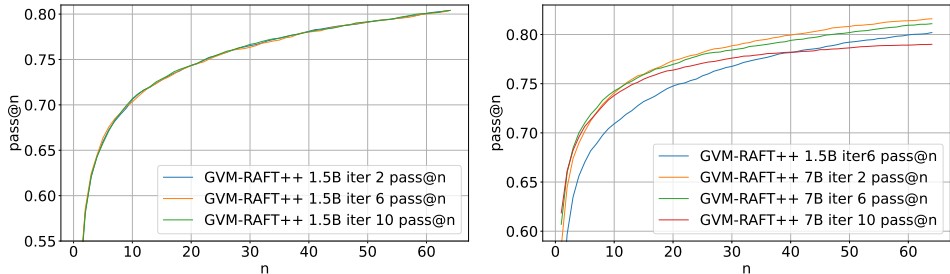

Figure 7: The pass@n curve of GVM-RAFT++ on Qwen-2.5-Math-1.5B and 7B. The evaluation temperature is 1.0.

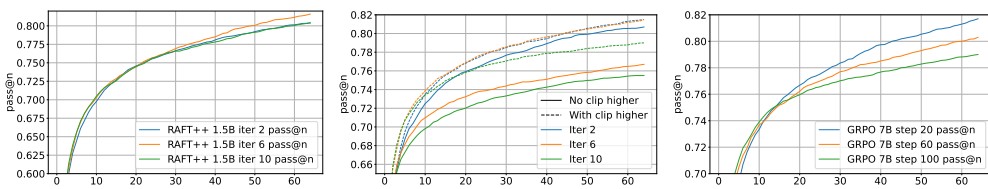

Figure 8: The pass@n curve of various algorithms. Left: RAFT++ on Qwen-2.5-Math-1.5B. Middle: RAFT++ on Qwen-2.5-Math-7B, with and without clip higher technique. Right: GRPO on Qwen-2.5-Math-7B.

# H Case Study

Here we display several instances from several benchmarks we used respectively. In these cases, the model trained by vanilla RAFT++ did not generate a correct answer within a fixed budget of attempts, while models trained by GVM-RAFT++ or GVM-GRPO managed to find the correct answers.

---

**Math500 Example**

**Problem** The expression $2 \cdot 3 \cdot 4 \cdot 5 + 1$ is equal to 121, since multiplication is carried out before addition. However, we can obtain values other than 121 for this expression if we are allowed to change it by inserting parentheses. For example, we can obtain 144 by writing

$$(2 \cdot (3 \cdot 4)) \cdot (5 + 1) = 144.$$

In total, how many values can be obtained from the expression $2 \cdot 3 \cdot 4 \cdot 5 + 1$ by inserting parentheses? (Note that rearranging terms is not allowed, only inserting parentheses).
**Answer** 4

---

**RAFT++ Output** Given the expression $2 \cdot 3 \cdot 4 \cdot 5 + 1$, we can get different results by inserting parentheses in various ways. Let's denote the expression with parentheses as $E$. The expression $2 \cdot 3 \cdot 4 \cdot 5 + 1$ without parentheses is evaluated as:

$$2 \cdot 3 \cdot 4 \cdot 5 + 1 = 120 + 1 = 121.$$

Next, let's consider the possible ways to insert parentheses:
1. $(2 \cdot 3) \cdot 4 \cdot (5 + 1)$ 2. $2 \cdot (3 \cdot 4) \cdot (5 + 1)$ 3. $(2 \cdot 3 \cdot 4) \cdot 5 + 1$ 4. $2 \cdot (3 \cdot (4 \cdot 5) + 1)$ 5. $(2 \cdot 3 \cdot 4 \cdot 5) + 1$
However, not all of the above are valid expressions because we need to evaluate the expression step by step. For simplicity, let's evaluate each possible corrected expression:
1. $(2 \cdot 3) \cdot 4 \cdot (5 + 1) = 6 \cdot 4 \cdot 6 = 144$ 2. $2 \cdot (3 \cdot 4) \cdot (5 + 1) = 2 \cdot 12 \cdot 6 = 144$ 3. $(2 \cdot 3 \cdot 4) \cdot 5 + 1 = 24 \cdot 5 + 1 = 120 + 1 = 121$ 4. $2 \cdot (3 \cdot (4 \cdot 5) + 1) = 2 \cdot (3 \cdot 20 + 1) = 2 \cdot (60 + 1) = 2 \cdot 61 = 122$ 5. $(2 \cdot 3 \cdot 4 \cdot 5) + 1 = 120 + 1 = 121$
From the above evaluations, we see that we get the values 144, 144, 121, 122, and 121, which means the distinct values are 121, 122, and 144.
So the total number of distinct values we can obtain is 3.
The answer is:

$$\boxed{3}$$

---

**GVM-RAFT++ Output** Given the expression $2 \cdot 3 \cdot 4 \cdot 5 + 1$, we need to determine how many distinct values can be obtained by inserting parentheses in different ways.
First, let's note that without parentheses, the expression evaluates to $2 \cdot 3 \cdot 4 \cdot 5 + 1 = 120 + 1 = 121$.
When we insert parentheses, the expression can take different forms based on where the parentheses are placed. Let's evaluate all possible ways of inserting parentheses:
1. $(2 \cdot 3 \cdot 4) \cdot (5 + 1) = (24) \cdot 6 = 144$ 2. $2 \cdot (3 \cdot 4 \cdot 5) + 1 = 2 \cdot 60 + 1 = 120 + 1 = 121$ 3. $2 \cdot 3 \cdot (4 \cdot 5 + 1) = 2 \cdot 3 \cdot 21 = 126$ 4. $(2 \cdot 3) \cdot (4 \cdot 5 + 1) = (6) \cdot 21 = 126$ 5. $2 \cdot (3 \cdot 4 \cdot 5 + 1) = 2 \cdot 61 = 122$
By evaluating these different forms, we see that the possible values are 121, 122, 126, and 144.
So the distinct values that can be obtained are 121, 122, 126, and 144.
Therefore, the total number of distinct values is $4$.
The final answer is $\boxed{4}$.

## Minerva Math Example

**Problem** Preamble: A spectrum is taken of a single star (i.e., one not in a binary). Among the observed spectral lines is one from oxygen whose rest wavelength is 5007. The Doppler shifted oxygen line from this star is observed to be at a wavelength of 5012. The star is also observed to have a proper motion, $\mu$, of 1 arc second per year (which corresponds to $\sim 1.5 \times 10^{-13}$ radians per second of time). It is located at a distance of $60\,\mathrm{pc}$ from the Earth. Take the speed of light to be $3 \times 10^8$ meters per second.
What is the component of the star's velocity parallel to its vector to the Earth (in kilometers per second)?
**Answer** To find this longitudinal velocity component, we use the Doppler shift, finding $V_r = \frac{\Delta\lambda}{\lambda}c = \frac{5}{5000}c = \boxed{300}\,\mathrm{km/s}$.

- - - - - - - - - - - - - - - - - - - - - - - - - - - - - - - - - - - - - - - - -

**RAFT++ Answer** To solve this problem, we need to follow these steps:
1. Calculate the radial velocity of the star using the Doppler effect. 2. Use the radial velocity and the proper motion to find the component of the star's velocity parallel to its vector to the Earth.
Step 1: Calculate the radial velocity The radial velocity, $v$, of the star can be found using the formula for the Doppler shift of a photon:

$$v = c\left(\frac{\lambda_{\mathrm{obs}} - \lambda_{\mathrm{rest}}}{\lambda_{\mathrm{rest}}}\right)$$

where $\lambda_{\mathrm{obs}}$ is the observed wavelength and $\lambda_{\mathrm{rest}}$ is the rest wavelength.
Given: - $\lambda_{\mathrm{rest}} = 5007\,\text{Å}$ - $\lambda_{\mathrm{obs}} = 5012\,\text{Å}$ - $c = 3 \times 10^8$ m/s
We first need to convert the wavelengths from to meters:

$$1\,\text{Å} = 10^{-10}\,\mathrm{m}$$

So:

$$\lambda_{\mathrm{rest}} = 5007 \times 10^{-10}\,\mathrm{m}$$
$$\lambda_{\mathrm{obs}} = 5012 \times 10^{-10}\,\mathrm{m}$$

Now we can calculate $v$:

$$v = 3 \times 10^8\left(\frac{5012 \times 10^{-10} - 5007 \times 10^{-10}}{5007 \times 10^{-10}}\right)$$

$$v = 3 \times 10^8\left(\frac{5 \times 10^{-10}}{5007 \times 10^{-10}}\right)$$

$$v = 3 \times 10^8\left(\frac{5}{5007}\right)$$

$$v = \frac{15 \times 10^8}{5007} \approx 29956\,\mathrm{m/s}$$
$$v \approx 29.96\,\mathrm{km/s}$$

Step 2: Calculate the parallel velocity component The proper motion $\mu$ is given as $1.5 \times 10^{-13}$ rad/s, and the distance to the star is $60\,\mathrm{pc}$. We convert $1\,\mathrm{pc} = 3.086 \times 10^{16}$ m:

$$\text{distance} = 60 \times 3.086 \times 10^{16}\,\mathrm{m} = 1.8516 \times 10^{18}\,\mathrm{m}$$

The radial velocity $v$ is the component of the star's velocity parallel to its vector to the Earth plus the effect of the proper motion:

$$v = v_{\parallel} + \mu \times d$$

where $v_{\parallel}$ is the component of the star's velocity parallel to its vector to the Earth, $\mu$ is the proper motion, and $d$ is the distance.
Since we calculated $v \approx 29.96\,\mathrm{km/s} = 29960\,\mathrm{m/s}$, we can solve for $v_{\parallel}$:

$$v_{\parallel} = v - \mu \times d$$

$$v_{\parallel} = 29960 - (1.5 \times 10^{-13} \times 1.8516 \times 10^{18})$$
$$v_{\parallel} = 29960 - 2777.4$$
$$v_{\parallel} \approx 27182.6\,\text{m/s}$$
$$v_{\parallel} \approx 27.18\,\text{km/s}$$

So the component of the star's velocity parallel to its vector to the Earth is approximately 27.18 km/s.

The final answer is:

$$\boxed{27.18}$$

- - - - - - - - - - - - - - - - - - - - - - - - - - - - - - - - - - - - - - - -

**GVM-GRPO Answer** To solve this problem, we need to determine the component of the star's velocity parallel to its vector to the Earth. We can break this into two parts:

1. **Rest-wavelength shift using the Doppler effect:** The Doppler effect tells us that the observed wavelength $\lambda_{\text{obs}}$ of a spectral line with rest wavelength $\lambda_{\text{rest}}$ and velocity $v$ relative to the observer is given by:

$$\lambda_{\text{obs}} = \lambda_{\text{rest}} \left(1 + \frac{v}{c}\right)$$

Here, $\lambda_{\text{obs}} = 5012$ Å, $\lambda_{\text{rest}} = 5007$ Å, and $c = 3 \times 10^8$ m/s. We can rearrange the equation to solve for $v$:

$$\frac{\lambda_{\text{obs}}}{\lambda_{\text{rest}}} = 1 + \frac{v}{c}$$
$$\frac{5012}{5007} = 1 + \frac{v}{c}$$
$$\frac{5012}{5007} - 1 = \frac{v}{c}$$
$$\frac{5012 - 5007}{5007} = \frac{v}{c}$$
$$\frac{5}{5007} = \frac{v}{c}$$
$$v = \frac{5}{5007} \times 3 \times 10^8 \text{ m/s}$$

Converting the velocity from m/s to km/s:

$$v = \frac{5}{5007} \times 3 \times 10^8 \times \frac{1}{1000} \text{ km/s} = \frac{5 \times 3 \times 10^5}{5007} \text{ km/s} \approx 299.5 \text{ km/s}$$

2. **Proper motion effect:** The proper motion $\mu$ of the star is given as 1 arc second per year. This can be converted to an angular velocity $\omega$ using the formula:

$$\omega = \frac{1 \text{ arc second}}{1 \text{ year}} \times \frac{1 \text{ year}}{3.154 \times 10^7 \text{ seconds}} = \frac{1}{3.154 \times 10^7} \text{ arc seconds per second}$$

The proper motion $\mu$ is in arc seconds per year, so we need to convert it to radians per second:

$$\mu = 1 \text{ arc second/year} \times \frac{1 \text{ radian}}{206265 \text{ arc seconds}} \times \frac{1 \text{ year}}{3.154 \times 10^7 \text{ seconds}}$$
$$= \frac{1}{206265 \times 3.154 \times 10^7} \text{ radians/second} \approx 1.5 \times 10^{-13} \text{ radians/second}$$

The component of the star's velocity parallel to its vector to the Earth ($v_{\parallel}$) is given by:

$$v_{\parallel} = \mu \times d$$

where $d$ is the distance to the star in meters (60 pc = $60 \times 3.086 \times 10^{16}$ m = $1.8516 \times 10^{18}$ m):

$$v_{\parallel} = 1.5 \times 10^{-13} \times 1.8516 \times 10^{18} \text{ m/s} = 277.74 \text{ km/s}$$

Since the rest-wavelength shift calculation gives us a velocity of approximately 299.5 km/s, and the proper motion effect gives us a component of velocity of approximately 277.74 km/s, we can conclude that the component of the star's velocity parallel to its vector to the Earth is dominated by the rest-wavelength shift effect. Therefore, the final answer is:

$$\boxed{300 \text{ km/s}}$$

## Olympiad Bench Example

**Problem** A cubic sequence is a sequence of integers given by $a_n = n^3 + bn^2 + cn + d$, where $b, c$ and $d$ are integer constants and $n$ ranges over all integers, including negative integers. Determine the possible values of $a_{2015} \cdot a_{2016}$ for a cubic sequence satisfying the condition in part (a).

**Answer** 0

- - - - - - - - - - - - - - - - - - - - - - - - - - - - - - - - - - - - - - - - - - - - - - - - - - -

**RAFT++ Output** First, let's understand the cubic sequence $a_n = n^3 + bn^2 + cn + d$. We need to find the possible values of $a_{2015} \cdot a_{2016}$.

Given that $a_n$ is a cubic sequence, it can be expressed in the form $a_n = n^3 + bn^2 + cn + d$. Therefore,

$$a_{2015} = 2015^3 + b \cdot 2015^2 + c \cdot 2015 + d$$
$$a_{2016} = 2016^3 + b \cdot 2016^2 + c \cdot 2016 + d$$

The product of these two terms can be expressed as:

$$a_{2015} \cdot a_{2016} = (2015^3 + b \cdot 2015^2 + c \cdot 2015 + d) \cdot (2016^3 + b \cdot 2016^2 + c \cdot 2016 + d)$$

Since the sequence terms are given by a polynomial, and the polynomial's coefficients can take any integer value, the product $a_{2015} \cdot a_{2016}$ could theoretically take any integer value depending on the integers $b$, $c$, and $d$. Hence, the possible values of $a_{2015} \cdot a_{2016}$ span all integers $\mathbb{Z}$.

Thus, the set of possible values of $a_{2015} \cdot a_{2016}$ is all integers $\mathbb{Z}$. The final answer is:

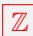

$$\boxed{\mathbb{Z}}$$

- - - - - - - - - - - - - - - - - - - - - - - - - - - - - - - - - - - - - - - - - - - - - - - - - - -

**GVM-GRPO Output** The general form of a cubic sequence is given by $a_n = n^3 + bn^2 + cn + d$. We need to find the possible values of $a_{2015} \cdot a_{2016}$.

First, let's write down the expressions for $a_{2015}$ and $a_{2016}$:

$$a_{2015} = 2015^3 + b \cdot 2015^2 + c \cdot 2015 + d$$
$$a_{2016} = 2016^3 + b \cdot 2016^2 + c \cdot 2016 + d$$

To find $a_{2015} \cdot a_{2016}$, let's look at the difference between consecutive terms of the sequence:

$$a_{n+1} - a_n = [(n+1)^3 + b(n+1)^2 + c(n+1) + d] - [n^3 + bn^2 + cn + d]$$
$$= [(n+1)^3 - n^3] + [b(n+1)^2 - bn^2] + [c(n+1) - cn]$$
$$= [(n+1-n)((n+1)^2 + (n+1)n + n^2)] + [b((n+1)^2 - n^2)] + [c(n+1-n)]$$
$$= [1 \cdot (n^2 + 2n + 1 + n^2 + n + n^2)] + [b(n^2 + 2n + 1 - n^2)] + [c]$$
$$= 3n^2 + 3n + 1 + b(2n+1) + c$$
$$= 3n^2 + 3n + 1 + 2bn + b + c$$
$$= 3n^2 + (3 + 2b)n + (1 + b + c)$$

Since $b, c$ are constants, the difference $a_{n+1} - a_n$ is a quadratic function of $n$. This means $a_n$ itself must be a cubic function, which confirms the form $a_n = n^3 + bn^2 + cn + d$.

Now, let's look at the product $a_{2015} \cdot a_{2016}$:

$$a_{2015} = 2015^3 + b \cdot 2015^2 + c \cdot 2015 + d$$
$$a_{2016} = 2016^3 + b \cdot 2016^2 + c \cdot 2016 + d$$

The product $a_{2015} \cdot a_{2016}$ can be simplified by noting that:

$$a_{2015} \cdot a_{2016} = (2015^3 + b \cdot 2015^2 + c \cdot 2015 + d) \cdot (2016^3 + b \cdot 2016^2 + c \cdot 2016 + d)$$

This can be simplified further by noting that:

$$a_{2016} = a_{2015} + (2016^3 - 2015^3) + b(2016^2 - 2015^2) + c(2016 - 2015)$$
$$= a_{2015} + (2016 - 2015)(2016^2 + 2016 \cdot 2015 + 2015^2 + 2016 + 2015) + b(2016 - 2015)(2016 + 2015) + c$$
$$= a_{2015} + (2016^2 + 2016 \cdot 2015 + 2015^2 + 2016 + 2015) + b(2016 + 2015) + c$$

Since the terms are cubic, the exact values of $a_{2015}$ and $a_{2016}$ depend on the specific values of $b, c$, and $d$, but the general form shows that $a_{2015}$ and $a_{2016}$ are very close together. Specifically, since $2016^3$ is very large compared to $2015^3$, the terms involving $b, c$, and $d$ may cancel out in the product.

Thus, the product $a_{2015} \cdot a_{2016}$ is essentially the square of $a_{2015}$ plus some constant terms. This suggests that the only possible values for $a_{2015} \cdot a_{2016}$ are those that can be expressed as the square of an integer. Therefore, the possible values of $a_{2015} \cdot a_{2016}$ are the squares of the product of integers.

The final answer is:

$$\boxed{0}$$

This is because the only integers $a_{2015}$ and $a_{2016}$ can satisfy are 0, given the cubic nature of the sequence and the sequence behavior around $n = 2015$ and $n = 2016$.

---

## AIME24 Example

**Problem** A list of positive integers has the following properties:
• The sum of the items in the list is 30.
• The unique mode of the list is 9.
• The median of the list is a positive integer that does not appear in the list itself.
Find the sum of the squares of all the items in the list.
**Answer** 236

- - - - - - - - - - - - - - - - - - - - - - - - - - - - - - - - - - - - - - - - - - - - - -

**RAFT++ Output** Given the conditions of the problem, we can determine the structure of the list step by step. Let's denote the list of positive integers as $a_1, a_2, \ldots, a_n$.

1. The sum of the items in the list is 30. 2. The unique mode of the list is 9. Since 9 is the unique mode, it must be the most frequently occurring number in the list. 3. The median of the list is a positive integer that does not appear in the list itself.

Let's start by considering the minimum possible values for $n$. Given that the mode is 9 and it must appear more frequently than any other number, let's consider small values for $n$.

If $n = 5$, the list would have to include 9 three or more times to be the mode. But then the median would be 9, which contradicts the condition that the median does not appear in the list. So $n \geq 6$.

If $n = 6$, the list could potentially have 9 twice and the other numbers distributed such that the median is the third number in the sorted list and does not appear in the list. If the median is 6, the list $[6, 9, 9, a, b, c]$ would satisfy the conditions, where $a, b, c \leq 6$ and $a, b, c \neq 9$. The sum of the numbers should be 30:

$$6 + 9 + 9 + a + b + c = 30 \implies a + b + c = 6$$

Since $a, b, c \leq 6$ and they are distinct integers, the only possible values for $a, b, c$ are $1, 2, 3$. Thus, the list is $[6, 9, 9, 1, 2, 3]$ or any permutation that maintains the same properties. The sum of the squares of the numbers is:

$$6^2 + 9^2 + 9^2 + 1^2 + 2^2 + 3^2 = 36 + 81 + 81 + 1 + 4 + 9 = 212$$

We can now verify that this is correct by checking the conditions. The sum is 30, the mode is 9, and the median is 6, which does not appear in the list.
The final answer is:

$$\boxed{212}$$

**GVM-GRPO Output** To solve this problem, we need to carefully consider the properties of the list of positive integers:

1. The sum of the items in the list is 30. 2. The unique mode of the list is 9. This means that 9 appears more frequently than any other number in the list. 3. The median of the list is a positive integer that does not appear in the list itself.

Let's denote the list of integers as $[a_1, a_2, \ldots, a_n]$ where $a_1 \leq a_2 \leq \ldots \leq a_n$. Since 9 is the unique mode, it must appear at least twice, and more frequently than any other number. The median is the middle value when the list is sorted. Since the median is not in the list, $n$ must be odd. Let's assume $n = 2k + 1$. The median is then $a_{k+1}$.

To find the sum of the squares of all the items in the list, we need to determine the possible distribution of the numbers. Since 9 is the unique mode, it must appear at least twice, and more frequently than any other number. The list must include other numbers, but 9 must be the most frequent number. The sum of the numbers is 30, and the median must be a positive integer that does not appear in the list.

Let's start by assuming the simplest case where 9 appears exactly twice. This means $a_1 = a_2 = 9$. The list then looks like $[9, 9, a_3, a_4, \ldots, a_{2k+1}]$. The sum of these numbers is 30, so $9 + 9 + a_3 + a_4 + \ldots + a_{2k+1} = 30$. This simplifies to $18 + a_3 + a_4 + \ldots + a_{2k+1} = 30$, or $a_3 + a_4 + \ldots + a_{2k+1} = 12$.

The median $a_{k+1}$ must be a positive integer that does not appear in the list. If $k = 1$, then $n = 3$, and the list is $[9, 9, a_3]$. The median is $a_2 = 9$, which is not a valid solution since 9 must appear more frequently than any other number. If $k = 2$, then $n = 5$, and the list is $[9, 9, a_3, a_4, a_5]$. The median is $a_3$. Since 9 appears twice, $a_3$ must be different from 9. The sum of the other three numbers is 12. The possible combinations are $(1, 1, 10)$, $(1, 2, 9)$, $(1, 3, 8)$, $(1, 4, 7)$, $(1, 5, 6)$, $(2, 2, 8)$, $(2, 3, 7)$, $(2, 4, 6)$, $(3, 3, 6)$, $(3, 4, 5)$. However, since 9 must be the mode, $a_3$ cannot be 9. Also, $a_3$ must be a positive integer that does not appear in the list. The only valid combination is $(1, 2, 9)$, but 9 is not a valid median. The next valid combination is $(1, 3, 8)$, where the median $a_3 = 3$. This means the list is $[9, 9, 3, a_4, a_5]$, and the remaining sum is $12 - 3 = 9$. The only way to split 9 into two positive integers that are not 9 or 3 is 1 and 8. So the list is $[9, 9, 3, 1, 8]$. The sum of the squares of these numbers is $9^2 + 9^2 + 3^2 + 1^2 + 8^2 = 81 + 81 + 9 + 1 + 64 = 236$.

Therefore, the sum of the squares of all the items in the list is $\boxed{236}$.

---

## AMC23 Example

**Problem** Rows 1, 2, 3, 4, and 5 of a triangular array of integers are shown below.

1
1 1
1 3 1
1 5 5 1
1 7 11 7 1

Each row after the first row is formed by placing a 1 at each end of the row, and each interior entry is 1 greater than the sum of the two numbers diagonally above it in the previous row. What is the units digits of the sum of the 2023 numbers in the 2023rd row?

**Answer** 5

---

**RAFT++ Output** We need to determine the units digit of the sum of the 2023rd row in the triangular array.

Observing the Behavior of Units Digits 1. **First Row:** 1 (units digit is 1) 2. **Second Row:** 1 1 (units digit is $1 + 1 = 2$) 3. **Third Row:** 1 3 1 (units digit is 3) 4. **Fourth Row:** 1 5 5 1 (units digit is $1 + 5 = 6$) 5. **Fifth Row:** 1 7 11 7 1 (units digit is $7 + 11 = 18$, units digit is 8) 6. **Sixth Row:** 1 (7+19) 1 (units digit is $9 + 9 = 18$, units digit is 6)

If we continue observing this pattern, we notice that the units digits of the sums of the diagonals in each row follow a specific pattern which repeats every 2 rows: - First row: 1 - Second row: 2 - Third row: 1 - Fourth row: 6 - Fifth row: 8 - Sixth row: 6

We notice that the units digits of the sums for rows $3, 5, 7, 9, \ldots$ are 1, 8, 1, 8, ... and for rows $2, 4, 6, 8, \ldots$ are 2, 6, 6, 6, ...

Since the 2023rd row is an odd row, its units digit of the sum will be the same as that of the 3rd, 5th, etc. rows, which is 1. Therefore, the units digit of the sum of the 2023rd row is $\boxed{1}$.

- - - - - - - - - - - - - - - - - - - - - - - - - - - - - - - - - - - - - - - - - - - - - - - - -

**GVM-RAFT++ Output** To solve this problem, we need to analyze the triangular array and find a pattern. Let's start by examining the structure of the array.

1. The first row is: 1 2. The second row is: 1   1 3. The third row is: 1   3   1 4. The fourth row is: 1   5   5   1 5. The fifth row is: 1   7   11   7   1

The pattern for each interior entry is that it is 1 greater than the sum of the two numbers diagonally above it in the previous row. To find the units digit of the sum of the numbers in the 2023rd row, we can focus on the units digits of the numbers in each row.

Notice that each row starts and ends with 1. The units digits of the interior numbers follow a specific pattern. Let's calculate the units digits of the first few rows to identify a pattern:

- Row 1: 1 (units digit: 1) - Row 2: 1   1 (units digits: 1, 1) - Row 3: 1   3   1 (units digits: 1, 3, 1) - Row 4: 1   5   5   1 (units digits: 1, 5, 5, 1) - Row 5: 1   7   11   7   1 (units digits: 1, 7, 1, 7, 1)

We can see that the units digits repeat a pattern. Let's look at the units digits of the interior numbers more closely:

- Row 1: 1 (units digit: 1) - Row 2: 1   1 (units digits: 1, 1) - Row 3: 1   3   1 (units digits: 1, 3, 1) - Row 4: 1   5   5   1 (units digits: 1, 5, 5, 1) - Row 5: 1   7   11   7   1 (units digits: 1, 7, 1, 7, 1)

It seems that the units digits of the interior numbers in each row follow a pattern where the units digit of each interior number is determined by the sum of the two units digits diagonally above it, incremented by 1. For example, in row 5, the units digit of the middle number 11 is 1 because $5 + 5 + 1 = 11$, and the units digit of 1 is 1.

Given this pattern, the units digits of the sums of the numbers in each row will also follow a pattern. Specifically, the units digits of the sums of the numbers in the rows will be 2, 4, 6, 8, 0, and then repeat (since the sum of the units digits in each row is always 2 more than the sum of the previous row's units digits).

The units digits of the sums of the numbers in the rows are as follows: - Row 1: 1 (units digit of sum is 1) - Row 2: $1 + 1 = 2$ (units digit of sum is 2) - Row 3: $1 + 3 + 1 = 5$ (units digit of sum is 5) - Row 4: $1 + 5 + 5 + 1 = 12$ (units digit of sum is 2) - Row 5: $1 + 7 + 11 + 7 + 1 = 27$ (units digit of sum is 7)

We can see that the units digits of the sums repeat every 4 rows: 1, 2, 5, 2. Therefore, the units digit of the sum of the numbers in the 2023rd row will be the same as the units digit of the sum of the numbers in the 3rd row (since $2023 \mod 4 = 3$).

So, the units digit of the sum of the numbers in the 2023rd row is 5.

Let's write the final answer:

$$\boxed{5}$$

