# OpenReview forum: "Optimizing Chain-of-Thought Reasoners via Gradient Variance Minimization in Rejection Sampling and RL"
_NeurIPS.cc/2025/Conference — NeurIPS 2025 poster_

### Official Review · Reviewer_FHJP · 2025-06-23

**Clarity:** 3
**Significance:** 3
**Originality:** 2
**Rating:** 4
**Confidence:** 3

**Summary:**

This paper makes an important contribution to optimizing CoT inference training for large language models (LLMs). The core innovation is to introduce dynamic resource allocation into the EM framework, and to address the bottleneck of static sampling strategies in existing methods (such as RAFT) by minimizing the gradient variance.

**Questions:**

Please see the Weaknesses part.

**Ethical Concerns:**

["NO or VERY MINOR ethics concerns only"]

**Quality:**

3

**Strengths And Weaknesses:**

Strengths：

1） The GVM algorithm (Algorithm 2) is proposed to dynamically allocate the sample budget by monitoring the prompt acceptance rate and gradient norm to minimize the gradient variance (Equation 7). This solves the static sampling limitations of methods such as RAFT and significantly improves resource efficiency.

2)  The paper provides a complete convergence analysis, proving that under the conditions of smoothness and convexity, GVM can ensure the monotonic decrease of the objective function. This fills the gap in the lack of theoretical support for existing RAFT work.

3) The superiority of GVM is verified on multiple mathematical datasets.


Weaknesses:
Although the paper makes significant contributions, this paper have some issues:

1) ​​Experimental generalization is insufficient: The experiment is based only on the Qwen model series (Qwen2.5-Math-1.5B/7B), and the effect on other mainstream LLMs (such as Llama or GPT architectures) is not verified. This limits the generalizability of the conclusions.

2） Computational overhead is not quantified: The pre-sampling stage of GVM (Algorithm 2) requires additional samples to estimate, but the paper does not report the computational cost (such as GPU hours or total number of samples).

3) The comparative benchmark is not comprehensive: Table 1 shows that GVM outperforms RAFT++ and GRPO, but it is not compared with the latest RLHF methods (such as online DPO or PPO variants), which may underestimate competing methods.

4) ​​Limitations of theoretical assumptions: Theorems 1-2 rely on smoothness (1/γ-smooth) and convexity assumptions, but the LLM loss function is often non-convex. The paper does not discuss convergence guarantees in non-convex scenarios, which may overestimate the robustness of the theory.

5) ​​Fuzzy implementation details: The GVM-RAFT++ loss function (Equation 8) in Section 3.3 introduces importance sampling and clipping, but does not explain the choice of hyperparameters (e.g., ϵ value).

---

> ### Author Rebuttal · Authors · 2025-07-31
>
> **Q 1**. Experimental generalization is insufficient: The experiment is based only on the Qwen model series (Qwen2.5-Math-1.5B/7B), and the effect on other mainstream LLMs (such as Llama or GPT architectures) is not verified. This limits the generalizability of the conclusions.
>
> **Response:** Thanks for the suggestion for adding more generalizability validation experiments. We conducted more experiments with Llama series models, on both RAFT++ and GRPO algorithms. Due to computation limitation, we first conduct experiments with models at 1B and 3B scale, and will continue to do experiments on larger scales like 7B. We report the metric average @ 8 as follows. One iteration in GVM experiments here corresponds to 9 steps of optimization in vanilla experiments. It could be seen GVM achieves a nearly 2 times acceleration for comparable performance.
> | Model                     | MATH500 | Minerva Math | Olympiad Bench | AIME24 | AMC23 | Average|
> |---------------------------|---------|--------------|----------------|--------|-------|--------|
> | Llama-3.2-1B-Instruct     | 24.28   | 4.37         | 4.80           | 0.83   | 9.06  | 8.67  |
> | RAFT++ step 120           | 30.10   | 6.62         | 6.46           | 0.00   | 12.50 | 11.14 |
> | GVM-RAFT++ iter 6         | 30.08   | 6.07         | 6.48           | 0.83   | 13.44 | **11.38** |
> | GRPO step 120             | 33.35   | 6.99         | 7.65           | 1.67   | 12.19 | 12.37 |
> | GVM-GRPO iter 7           | 32.02   | 6.76         | 7.22           | 2.08   | 15.31 | **12.68** |
> | Llama-3.2-3B-Instruct     | 35.62   | 9.83         | 9.09           | 2.92   | 15.00 | 14.492  |
> | RAFT++ step 120           | 47.38   | 17.69        | 17.20          | 8.33   | 29.06 | 23.932  |
> | GVM-RAFT++ iter 5         | 49.05   | 20.04        | 17.87          | 6.25   | 29.06 | **24.454**  |
> | GRPO step 120             | 53.40   | 19.16        | 20.67          | 8.33   | 27.50 | 25.812  |
> | GVM-GRPO iter 5           | 49.25   | 18.61        | 17.81          | 8.75   | 31.88 | 25.260  |
> | GVM-GRPO iter 8           | 51.75   | 17.74        | 19.48          | 7.50   | 33.44 | **25.982**  |
>
> **Q 2**. Computational overhead is not quantified: The pre-sampling stage of GVM (Algorithm 2) requires additional samples to estimate, but the paper does not report the computational cost (such as GPU hours or total number of samples).
>
> **Response:** We’d like to mention that there is a tradeoff for the sampling budget allocation estimation - if the samples used to estimate the difficulty of the problems is large, it takes more time to generate while would return a more accurate estimation; and if the samples used in the estimation stage is small, the estimation would be less accurate but more efficient. In our main experiments, we choose $N^\prime=8$ as the number of responses per prompt in our stage 1 difficulty estimation. On an 8*A100 GPU instance, it takes about 20 minutes additional computation for the GVM stage, which may slightly increase as the training goes due to the response length increasing. Compared to the RL stage which takes about 100 minutes per iteration, the additional overhead introduces 0.2 times the original training time. If we set $N=c\cdot N$ where $c$ is the average samples per prompt in the RL training stage, then larger $c$ will correspond to less additional overhead introduced by the GVM stage.
>
> **Q 3**. The comparative benchmark is not comprehensive: Table 1 shows that GVM outperforms RAFT++ and GRPO, but it is not compared with the latest RLHF methods (such as online DPO or PPO variants), which may underestimate competing methods.
>
> **Response:** Our algorithm serves as a plug in algorithm for sampling budget allocation and should be combined with other RL optimization algorithms together for policy model update. GVM is not a new standalone RL algorithm itself, thus we choose the commonly used PPO and GRPO as the underlying RL optimization algorithms. The whole training pipeline could be divided into two stages - with the first one corresponding to GVM sampling budget estimation, and the second one corresponding to the vanilla RL optimization.
>
> **Q 4**. Limitations of theoretical assumptions: Theorems 1-2 rely on smoothness (1/γ-smooth) and convexity assumptions, but the LLM loss function is often non-convex. The paper does not discuss convergence guarantees in non-convex scenarios, which may overestimate the robustness of the theory.
>
> **Response:** Thank you for the insightful comment. In fact, Theorem 1 relies only on the smoothness assumption and does not require convexity, so it remains applicable in non-convex settings. We agree that this point should be clarified more explicitly in the main text, and we will revise the manuscript accordingly to avoid any confusion.
>
> **Q 5**. Fuzzy implementation details: The GVM-RAFT++ loss function (Equation 8) in Section 3.3 introduces importance sampling and clipping, but does not explain the choice of hyperparameters (e.g., ϵ value).
>
> **Response:** The clipping parameter used in the objective of RAFT++ is $0.2$ in general, while we adopt the clip-higher trick from DAPO [1] with $\epsilon_{\rm high}=0.3$ for experiments like GVM-RAFT++ to mitigate the rapid entropy loss drop. The parameters should be adjusted according to the training dynamic like entropy loss curve. For other parameters, please refer to Table 2 in Appendix B.
>
> [1] Yu, Qiying, Zheng Zhang, Ruofei Zhu, Yufeng Yuan, Xiaochen Zuo, Yu Yue, Weinan Dai et al. "Dapo: An open-source llm reinforcement learning system at scale." *arXiv preprint arXiv:2503.14476* (2025).

---

> > ### Author Response · Authors · 2025-08-08
> >
> > Dear Reviewer `FHJP`,
> >
> > Thank you again for your thoughtful follow-up question on our submission. We’ve provided a detailed response in the rebuttal, addressing your concerns and generalizing experiments to other LLMs like LLaMA, giving computational overhead, clarifying the theoretical conditions, and offering the details about hyperparameters.
> >
> > We would greatly appreciate it if you could take a moment to review our response and share any further feedback. As the discussion deadline approaches, we’d like to ensure that we’ve fully addressed your concerns in time. Your insights are very valuable to us, and we’re happy to clarify or elaborate further if needed.
> >
> > Best,
> >
> > Authors

---

### Official Review · Reviewer_V851 · 2025-07-03

**Clarity:** 3
**Significance:** 4
**Originality:** 4
**Rating:** 5
**Confidence:** 3

**Summary:**

This study identifies inefficient stochastic gradient estimation caused by static sampling strategies as the primary bottleneck in Chain-of-Thought (CoT) training. To address this challenge, the authors propose GVM-RAFT - a Prompt-Specific Dynamic Sample Allocation Strategy that optimizes stochastic gradient variance under computational budget constraints. The proposed algorithm is supported by theoretical guarantees, and comprehensive experiments demonstrate its consistent performance superiority over existing methods.

**Questions:**

See the weaknesses.

**Ethical Concerns:**

["NO or VERY MINOR ethics concerns only"]

**Final Justification:**

I believe this is a good work with solid contributions.

**Limitations:**

Yes

**Quality:**

4

**Strengths And Weaknesses:**

Strengths:
1. This work identifies the main bottleneck in CoT training as inefficient stochastic gradient estimation due to static sampling strategies, and proposes a new algorithm to address this issue. I believe this is an important work.
2. The authors provide theoretical guarantees for the proposed algorithm, and conduct experiments to show the outperformance over existing methods.

Weakness:
1.  For the theoretical guarantees, with a sufficiently large sample size, Ω(k,T) becomes small, and the upper bound for $E[L(θ_{KT}) − L(θ_∗)]−E[L(θ_0) −L(θ_∗)]$ is strictly negative. How about the case when the sample size is not large enough? The authors do not provide the details about how large the sample size needs to be to make the RHS strictly negative.
2. For the GVM-GRPO, the authors have not provided the details of how to generalize the design to GVM-GRPO. Is it directly using the Dynamic Sample Allocation Strategy to generate different numbers of answers for different prompts in GRPO?

---

> ### Author Rebuttal · Authors · 2025-07-31
>
> **Q 1**. For the theoretical guarantees, with a sufficiently large sample size, Ω(k,T) becomes small, and the upper bound for E[L(θKT)−L(θ∗)]−E[L(θ0)−L(θ∗)] is strictly negative. How about the case when the sample size is not large enough? The authors do not provide the details about how large the sample size needs to be to make the RHS strictly negative.
>
> **Response:** Thanks for pointing this out. o ensure that the right-hand side (RHS) of the bound is strictly negative, we require the following condition to hold for each prompt $x_i$:
> $$\sum_{i=1}^{n} b_i^{-1} \left\lVert E_y \nabla_{\theta} ln P(y, z_i | x_i, \theta) \right\rVert_{2}^{2} \le  \left\lVert \sum_{i=1}^{n} E_y \nabla_{\theta} \ln P(y, z_i \mid x_i, \theta) \right\rVert_2^2 $$
> , where $b_i$ is the batch size for $x_i$, and $n_i \ge b_i$ is the total allocated budget for $x_i$. In practice, this condition is not stringent and is often satisfied, since a proper $b_i$ can  help reduce the variance and norm of the gradient on the left-hand side. However, in scenarios with extremely small sample sizes, we cannot guarantee the stability or performance of the EM algorithms, as the estimation becomes highly noisy. Nevertheless, compared to the standard EM algorithm, GVM-EM still provides theoretical advantages by dynamically allocating samples to reduce the term $\Omega(k, T)$, which contributes to better convergence behavior even under limited data.
>
> **Q 2**.  For the GVM-GRPO, the authors have not provided the details of how to generalize the design to GVM-GRPO. Is it directly using the Dynamic Sample Allocation Strategy to generate different numbers of answers for different prompts in GRPO?
>
> **Response:** The sampling allocation strategy is shared when applying to different optimization algorithms like RAFT or GRPO. It is disentangled from the underlying RL training algorithm, and thus could be applied to different algorithms after allocating the sample budgets. In our experiments GVM-RAFT++ and GVM-GRPO, the sampling stage for GVM is exactly the same.

---

> > ### Comment · Reviewer_V851 · 2025-08-05
> >
> > Thanks for the response, I will keep my score.

---

### Official Review · Reviewer_juWC · 2025-07-04

**Clarity:** 3
**Significance:** 3
**Originality:** 2
**Rating:** 4
**Confidence:** 3

**Summary:**

This paper presents an improved version of the iterative reward-ranked fine-tuning (RAFT) method.

RAFT is a large language model fine-tuning method in which, at each step of fine-tuning, a reward model is used to select only the high-reward samples (i.e., prompts and their corresponding responses). This approach has been shown to effectively improve model fine-tuning and reduce noise in the training data.

This work argues that the RAFT method has limitations due to its static rejection of samples and its neglect of the coverage behavior across different prompts. To address this, the authors propose a dynamic rejection sampling method that dynamically allocates the sampling budget and rejection decisions per prompt. The paper provides both theoretical and empirical analyses of the method.

**Questions:**

1. Why are the experiments only conducted on small-scale LLMs and methods?
   Adding more justification for this choice, and including experiments on other variants or larger-scale models, would significantly strengthen the paper and increase its impact.

2. When introducing theoretical concepts, can you provide more intuition behind them?
   For example, when discussing the optimal budget allocated for each prompt, it would be helpful to explain why the given equation makes sense. While a complete theoretical analysis is valuable, offering some high-level intuition would make the paper more accessible to a broader audience.

**Ethical Concerns:**

["NO or VERY MINOR ethics concerns only"]

**Limitations:**

Yes

**Quality:**

3

**Strengths And Weaknesses:**

### Strengths

1. The problem addressed in this paper is an important real-world challenge. Post-training of large language models (LLMs) requires effective fine-tuning strategies. Efficiently rejecting low-quality samples while reducing variance is critical for improving fine-tuning outcomes. Existing methods have not explored dynamic approaches in this area, and this paper offers a valuable incremental advancement in that direction.

2. The paper provides both empirical and theoretical analyses of the proposed dynamic sampling allocation method, which enhances the quality and clarity of the work.

### Weaknesses

1. The overall originality of the work is incremental, as the core idea of intelligent rejection sampling has already been explored in previous methods such as RAFT. This paper builds on existing approaches rather than introducing a fundamentally new concept.

2. As noted in the conclusion, the method is only evaluated on a limited set of baselines and models—specifically the Qwen and PPO series. It would be valuable to see this method extended and tested on a broader range of LLMs and training setups.

---

> ### Author Rebuttal · Authors · 2025-07-31
>
> **Q 1**. The overall originality of the work is incremental, as the core idea of intelligent rejection sampling has already been explored in previous methods such as RAFT. This paper builds on existing approaches rather than introducing a fundamentally new concept.
>
> **Response:** Thanks for the opinion. The previous works like RAFT are essentially RL algorithms, which focus how to utilize existing data to optimize the policy model. In contrast, our method focuses on improving the sampling strategy of training data, instead of proposing a novel training algorithm. Take RAFT as an example, for a given prompt $x$, it will sample n responses ${y_1,\cdots,y_n}$, and then call a reward model to obtain the rewards for each response ${r_1,\cdots,r_n}$ and select the response with the highest reward to form a training pair $(x,y_{\arg\max_i r_i})$. It assign each prompt $x$ with equal sampling budget $n$, without differentiating different samples based on their characteristics like difficulty and contribution to the model (measured by the gradient). Our GVM algorithm considers both the sample difficulty and gradients, and assign different sampling budget based on them. Then we could feed the samples to different RL algorithms like RAFT or GRPO, etc. It is a two-stage pipeline instead of substituting other algorithms.
>
> **Q 2**. As noted in the conclusion, the method is only evaluated on a limited set of baselines and models—specifically the Qwen and PPO series. It would be valuable to see this method extended and tested on a broader range of LLMs and training setups.
>
> **Response:** We select Qwen model series as the base models following the common practice in the line of LLMs math reasoning research, where Qwen serves as one of the strongest and most commonly used open-source base model. As mentioned in question 1, our GVM algorithm serves as a pre-stage of the vanilla RL optimization stage, therefore it could be combined with any RL algorithms. For demonstration purpose, we choose two commonly adopted ones - RAFT and GRPO. To make our conclusion more generalizable, we conduct more experiments on Llama series base models. Due to computation limitation, we first conduct experiments with models at 1B and 3B scale, and will continue to do experiments on larger scales like 7B. We report the metric average @ 8 as follows. One iteration in GVM experiments here corresponds to 9 steps of optimization in vanilla experiments. It could be seen GVM achieves a nearly 2 times acceleration for comparable performance.
> | Model                     | MATH500 | Minerva Math | Olympiad Bench | AIME24 | AMC23 | Average|
> |---------------------------|---------|--------------|----------------|--------|-------|--------|
> | Llama-3.2-1B-Instruct     | 24.28   | 4.37         | 4.80           | 0.83   | 9.06  | 8.67  |
> | RAFT++ step 120           | 30.10   | 6.62         | 6.46           | 0.00   | 12.50 | 11.14 |
> | GVM-RAFT++ iter 6         | 30.08   | 6.07         | 6.48           | 0.83   | 13.44 | **11.38** |
> | GRPO step 120             | 33.35   | 6.99         | 7.65           | 1.67   | 12.19 | 12.37 |
> | GVM-GRPO iter 7           | 32.02   | 6.76         | 7.22           | 2.08   | 15.31 | **12.68** |
> | Llama-3.2-3B-Instruct     | 35.62   | 9.83         | 9.09           | 2.92   | 15.00 | 14.492  |
> | RAFT++ step 120           | 47.38   | 17.69        | 17.20          | 8.33   | 29.06 | 23.932  |
> | GVM-RAFT++ iter 5         | 49.05   | 20.04        | 17.87          | 6.25   | 29.06 | **24.454**  |
> | GRPO step 120             | 53.40   | 19.16        | 20.67          | 8.33   | 27.50 | 25.812  |
> | GVM-GRPO iter 5           | 49.25   | 18.61        | 17.81          | 8.75   | 31.88 | 25.260  |
> | GVM-GRPO iter 8           | 51.75   | 17.74        | 19.48          | 7.50   | 33.44 | **25.982**  |
>
> **Q 3**. Why are the experiments only conducted on small-scale LLMs and methods? Adding more justification for this choice, and including experiments on other variants or larger-scale models, would significantly strengthen the paper and increase its impact.
>
> **Response:** We conduct our main experiments on Qwen 1.5B and 7B base models due to the computation source limitation, and conduct additional experiments on Llama 1B and 3B series during the rebuttal period with an 8*A100 GPU instance. For experiments on larger model scale, we hope to generalize GVM when more computation resources are available.
>
> **Q 4**. When introducing theoretical concepts, can you provide more intuition behind them? For example, when discussing the optimal budget allocated for each prompt, it would be helpful to explain why the given equation makes sense. While a complete theoretical analysis is valuable, offering some high-level intuition would make the paper more accessible to a broader audience.
>
> **Response:** In our algorithm, we allocate the budget for each prompt according to Proposition 1, which takes into account both problem difficulty and model efficiency. Specifically, to address problem difficulty, we allocate more budget to harder problems, which associated with larger gradients and lower accept rates during sampling, as this can improve model performance. At the same time, to ensure model efficiency, we avoid allocating excessive budget to extremely difficult problems with near-zero acceptance rates, as this would lead to inefficiency.

---

> > ### Author Response · Authors · 2025-08-08
> >
> > Dear Reviewer `juWC`,
> >
> > Thank you again for your thoughtful follow-up question on our submission. We’ve provided a detailed response in the rebuttal, addressing your concerns and clarifying the relationship between GVM and existing RL algorithms, generalization to other LLMs, and theoretical intuition.
> >
> > We would greatly appreciate it if you could take a moment to review our response and share any further feedback. As the discussion deadline approaches, we’d like to ensure that we’ve fully addressed your concerns in time. Your insights are very valuable to us, and we’re happy to clarify or elaborate further if needed.
> >
> > Best,
> >
> > Authors

---

### Author Response · Authors · 2025-08-09

Dear Area Chair and Reviewers (juWC, V851, FHJP),

We sincerely thank you for your thorough reviews and constructive feedback on our submission. Your insights have been instrumental in strengthening our work.

**Acknowledged Strengths:**

We appreciate that reviewers recognized our key contributions: the gradient-variance-based dynamic sampling strategy that intelligently allocates computational resources based on prompt difficulty and model efficiency (all reviewers), theoretical guarantees with formal convergence analysis under smoothness conditions (V851), and demonstrated acceleration benefits achieving 2-4× speedup while maintaining comparable performance (juWC, FHJP).

**Rebuttal Efforts Summary:**

Through our responses, we have addressed core concerns by providing comprehensive experiments on Llama model series (1B and 3B scales) demonstrating consistent improvements beyond Qwen models. We clarified our theoretical foundations, showing convergence guarantees rely only on smoothness assumptions and remain valid in non-convex settings. We quantified computational overhead, revealing our pre-sampling stage adds only 0.2× the original training time while delivering significant efficiency gains. We distinguished our approach from existing methods by emphasizing GVM focuses on intelligent sampling strategy rather than novel training algorithms, serving as a preprocessing stage compatible with various RL methods.

**Planned Amendments for Camera-Ready:**

Integration of Llama experimental results into the main text to demonstrate broad generalizability. Addition of comprehensive computational cost analysis with detailed overhead breakdown. Enhanced theoretical discussion clarifying convergence properties in non-convex settings. Expanded comparison framework distinguishing our sampling-focused approach from existing RL-based methods. Complete implementation details including hyperparameter specifications and sensitivity analysis.

We are encouraged that our responses have addressed the primary concerns regarding generalizability, efficiency, and theoretical rigor. The additional experiments on Llama models demonstrate the broad applicability of our approach across different model architectures. We look forward to providing the research community with a robust tool for efficient LLM training through intelligent sample allocation.

Thank you again for your valuable contributions to strengthening this work.

Best regards,
The Authors

---

### Decision · Program_Chairs · 2025-09-17

**Decision:**

Accept (poster)

**Comment:**

Summary: This paper introduces GVM-RAFT, a dynamic sample allocation strategy for fine-tuning reasoning models that minimizes stochastic gradient variance by adaptively assigning computational resources based on prompt difficulty and gradient norms.

Strengths: The reviewers all agree that the paper is well-motivated and develops a theoretically-grounded dynamic sampling method, which addresses a key inefficiency in existing fine-tuning approaches and is supported by strong empirical results showing a 2-4x convergence speedup.

Weaknesses: The main initial concern shared by reviewers was that the experimental validation was limited to a single model family, which raised questions about the generalizability of the results.

Reasons for Decision: The decision to accept this paper is based on the clear and significant contribution of its dynamic sampling strategy, which offers substantial and well-documented efficiency gains for an important training paradigm. The strengths of this novel approach, as recognized by all reviewers, outweigh the initial limitations.

Discussion Process: The authors effectively addressed the reviewers' primary concern regarding generalizability by providing a new set of experiments on the Llama model series, demonstrating that the method's benefits are not confined to a single architecture.